# Study on Phase Characteristics of Wind Pressure Fields around a Prism Using Complex Proper Orthogonal Decomposition

**Tomoyuki Murakami [1,2,*], Yuichiro Nishida [2] and Tetsuro Taniguchi [2]**

1   General Building Research Corporation of Japan, Osaka 565-0873, Japan
2   Graduate School of Engineering, Osaka City University, Osaka 558-8585, Japan
*   Correspondence: to-murakami@gbrc.or.jp; Tel.: +81-6-834-0919

**Abstract:** Wind loads for the design of wind-resistant high-rise buildings are generally evaluated based on spectral modal analysis or time-history response analysis using wind pressure data obtained from wind tunnel experiments with rigid models. The characteristics of the fluctuating wind pressures around vibrating buildings must be evaluated for relevant wind-resistant designs because the wind pressures around buildings are affected by their vibrations. One of the methods to investigate fluctuating fields is complex proper orthogonal decomposition (CPOD), which can express complicated pressure fields, including advection phenomena, as coherent structures. This paper presents the phase characteristics of fluctuating wind pressures around rigid and elastic models of a square-sectioned prism evaluated via CPOD analysis using the results of wind tunnel experiments. The evaluation procedure for the symmetricity of the fluctuating wind pressure modes obtained via CPOD is presented. The similarity of fluctuating wind pressure fields is evaluated as the congruency of the planes formed by the 1st- and 2nd-eigenmodes. With symmetricity and similarity, the fluctuating wind pressure fields are classified into three types: resonant and non-resonant states in smooth flow, and in gradient flow. The characteristics of the three types of wind pressure fields are shown, respectively, in the symmetric and anti-symmetric modes.

**Keywords:** fluctuating wind pressure field; complex proper orthogonal decomposition; prism; wind tunnel experiment; eigenmode; rigid model; elastic model; vibration; phase characteristics; high-rise building

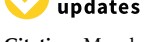



## 1. Introduction

For a rational design of wind-resistant high-rise buildings, it is crucial to correctly estimate the wind pressures around them. Accordingly, several researchers have estimated the wind pressures around high-rise buildings [1–8]. In general, the wind pressures are evaluated through wind tunnel experiments with rigid models. The characteristics of wind pressures must be studied to consider building vibrations because they can be affected by the vibration of buildings. Additionally, proper orthogonal decomposition (POD) has been frequently used as a method for detecting the coherent structures of fluctuating wind pressure fields when studying the wind pressures around buildings.

Tamura [9] validated POD by evaluating the fluctuating wind pressure fields around low- and high-rise building models using it. Li [10] also evaluated the eigenvalues and contribution ratio of POD for the wind pressures measured on the sloped roof surfaces of a low-rise building in a wind tunnel experiment to demonstrate the validity and usefulness of the method. Kim [11] conducted a wind tunnel experiment on two linked super- high-rise buildings and used POD analysis to evaluate the fluctuating wind pressure fields around the two building models. Jeong [12] used POD to analyse the fluctuating wind pressure fields on the roof surfaces of a low-rise building model in occurrences where the wind pressure measurement holes were both uniformly and non-uniformly placed. Cao [13] used POD to evaluate the peak pressure of a high-rise building at two localized regions with

severe suction: the lower windward corner of the side wall and the region immediately upstream of the trailing edge. Zhou [14] used POD to investigate the effect of gap distance on the aerodynamic interference effect of two neighbouring buildings. Mohammadi [15] and Bourgeois [16] analysed the velocity fields in the wake of a square cylinder using POD for the results of a wind tunnel experiment. Several other studies using POD analysis have been conducted on buildings and civil engineering structures [17–25].

Complex proper orthogonal decomposition (CPOD) [26,27], which is an extension of POD, has also been used in numerous studies. The CPOD method is based on the idea of the analytic signal presented by Gabor [28], which is obtained by excluding the negative frequency components in a real signal and doubling the positive frequency components. The concept of the analytic signal is used in various fields and is crucial in CPOD. Taniguchi [29] showed that advection phenomena of wind pressures can be expressed as a single principal component by CPOD. This is a general method that includes POD expressing a fluctuating field as modes fixed in space. The CPOD method was also applied to standing and traveling wave examples to demonstrate its effectiveness. Chauve [30] performed a CPOD analysis of the flow field around a row of cylinders. The coefficients of the thermodynamic Ginzburg–Landau type equation were estimated using the eigenmodes discovered in the study. Harlander [31] conducted a CPOD analysis on the flow of a thermally driven heating rotor measured by particle image velocimetry and laser doppler velocimetry to discuss the flow fields. Pfeffer [32] performed a CPOD analysis on three experimental data sets for flows over a two-wave sinusoidal bottom profile in a thermally driven rotating annular fluid. The results of CPOD analysis provide a vivid description of the wave behaviour. Furthermore, several studies have used CPOD in fields other than architecture or civil engineering [33–35].

As described above, CPOD has been used to investigate various flow fields around buildings and is considered as an effective method for such cases. Although Nakamura [36] investigated the pressure fields around a flat plate roof, few studies have used CPOD on fluctuating wind pressure fields around buildings, including the case in which the building is vibrating. As the characteristics of fluctuating wind pressure fields can be clarified by CPOD, this method will provide valuable knowledge for the wind-resistant design of high-rise buildings.

The purpose of this study was to evaluate the phase characteristics of fluctuating wind pressure fields around a prism and provide valuable knowledge for the wind-resistant design of high-rise buildings. This paper consists of five sections. In Section 2, the set-up of the wind tunnel experiments, measurement procedures, rigid and elastic models of a square-sectioned prism, experimental parameters, and results of experiments are described. The wind tunnel experiments were performed in both smooth and gradient flows, measuring the pressures around the models and the displacement of the elastic model. The wind pressure coefficients and displacements were determined by the experiments. Furthermore, the displacement calculated according to the equation of motion using the wind forces acting on the rigid and elastic models is presented. In Section 3, the CPOD analysis procedure is explained. The pressure fields around the rigid and elastic models were investigated using the characteristics of the eigenmodes and principal coordinates using CPOD. In Section 4, the calculation procedure for the symmetricity of the eigenmode is presented. The calculation procedure for the similarity of wind pressure fields as the congruency of the plane formed by the two principal eigenmodes is also shown. Based on these results, the pressure fields around the models were classified into three types according to their flow characteristics and responses. In Section 5, the characteristics of the pressure field for the types classified in Section 4 are discussed.

## 2. Wind Tunnel Experiment

### 2.1. Wind Tunnel Flow

Wind tunnel experiments were conducted in an Eiffel-type wind tunnel at the General Building Research Corporation of Japan. The wind tunnel is shown in Figure 1. This study

used two types of wind tunnel flows: a smooth flow and a gradient flow. The smooth flow had a uniform vertical wind speed profile and little turbulence, and the gradient flow was in a turbulent boundary layer generated on the wind tunnel floor by a spire and roughness blocks. The power-law exponent of the gradient flow, $\alpha$, for the mean wind-speed profile was approximately 0.20. The profiles of the mean wind speed ratio $U_z/U_{600}$ and turbulence intensity $I_u$ above the centre of the turntable in the wind tunnel are shown in Figure 2, where $U_z$ is the wind speed at a height of $z$ mm. Figure 3a shows the dimensionless power spectral density (PSD), $fS_u(f)/\sigma^2$, of the gradient flow at the reference height $H$ ($z$ = 375 mm), where $f$ is the frequency, $S_u(f)$ is the power spectrum of the fluctuating wind speed, and $\sigma^2$ is the variance of the wind speed. The general shape of the PSD is consistent with the Karman-type spectrum with $L_z$ = 0.41 m, which corresponds to 164 m when multiplied by the scale factor of the model (=1/400, as described later in this section). Figure 3b shows a comparison of the turbulence scale (approximately 224 m) between the wind tunnel gradient flow and the value in the AIJ Recommendations for Loads on Buildings [37] (hereafter referred to as the AIJ-RLB). Comparing the value of $L_z$ with the recommended value in the AIJ-RLB for suburban exposure (Terrain Category III), $L_z$ seems to be smaller but is within the range of variability of the observations.

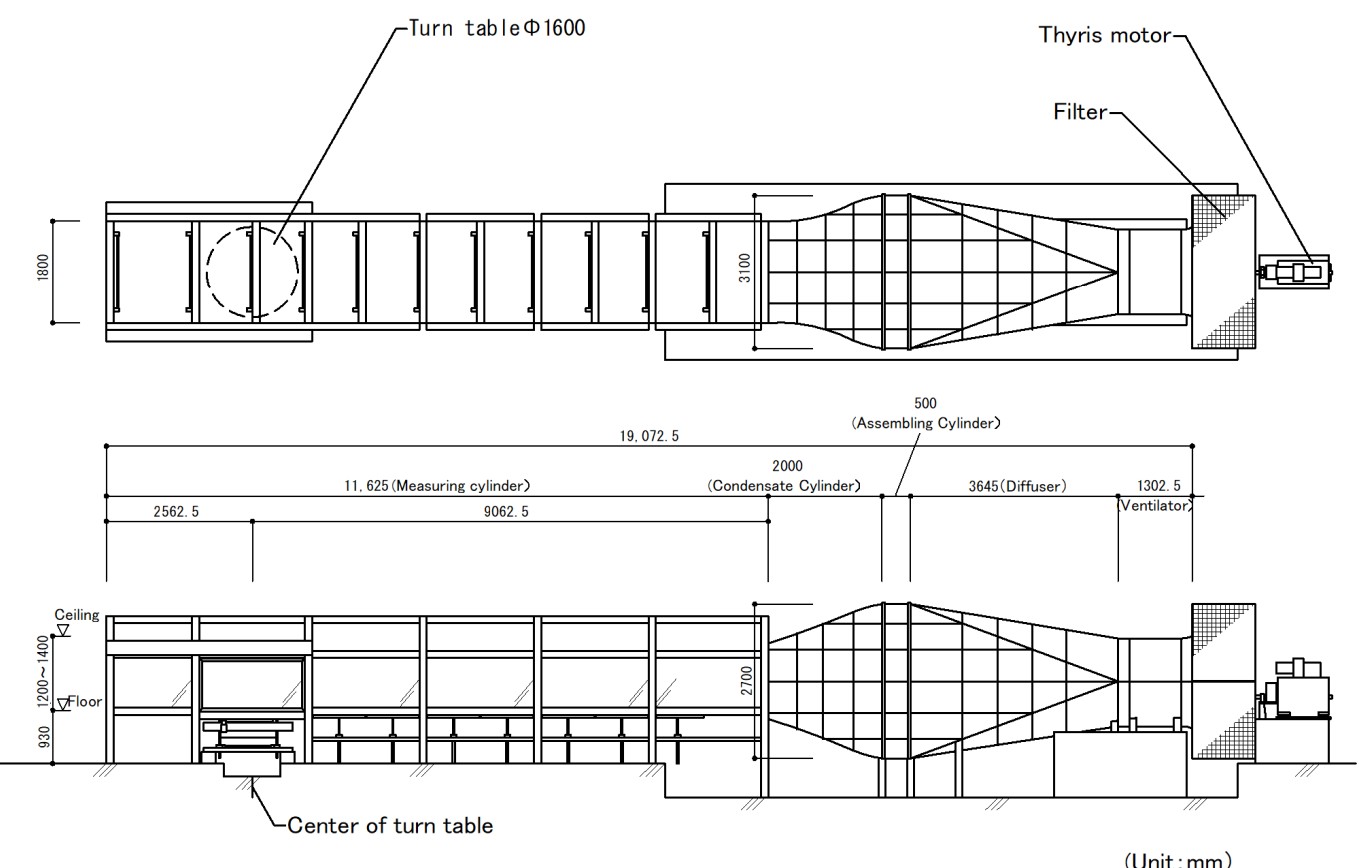

**Figure 1.** Wind Tunnel.

## 2.2. Experiments Using a Rigid Model

The target of this study was a high-rise building that was 30 m wide × 30 m deep × 150 m high. The model scale was set at 1/400, and the experimental rigid model was a square-sectioned prism 75 mm wide × 75 mm deep × 375 mm high. An overview of the rigid model is presented in Figure 4. The arrangement of the wind-pressure measurement points is shown in Figure 5. In total, 120 wind pressure measurement points were established with 30 points on each side of the model. A brass tube (40 mm in length, 0.8 mm in inner diameter) and a vinyl tube (1000 mm in length, 1.0 mm in inner diameter) were attached to the wind pressure measurement hole (0.8 mm in diameter). The rigid model

was made of acrylic material, and vinyl tubing was led from the inside to the bottom of the model.

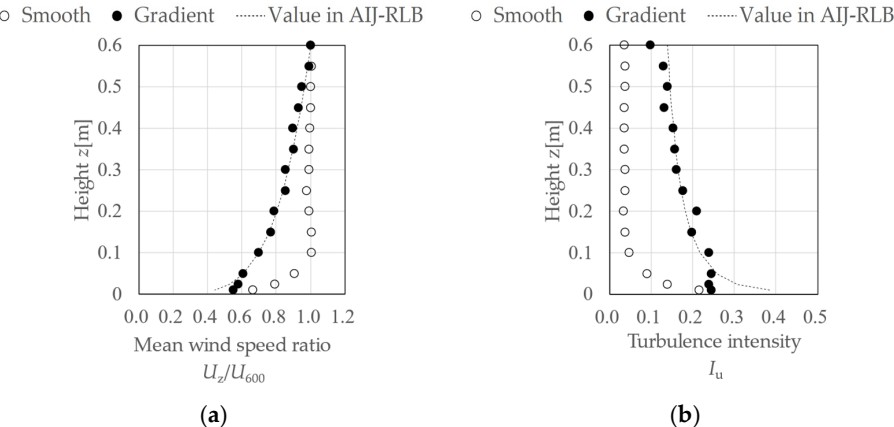

**Figure 2.** Profiles of wind tunnel flow: (**a**) mean wind speed ratio; (**b**) turbulence intensity.

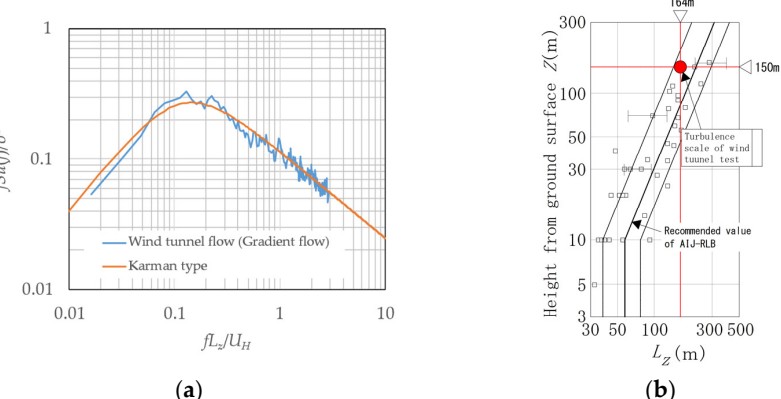

**Figure 3.** Characteristics of gradient flow: (**a**) PSD at the height of $z$ = 375 mm; (**b**) turbulence scale observation in AIJ-RLB.

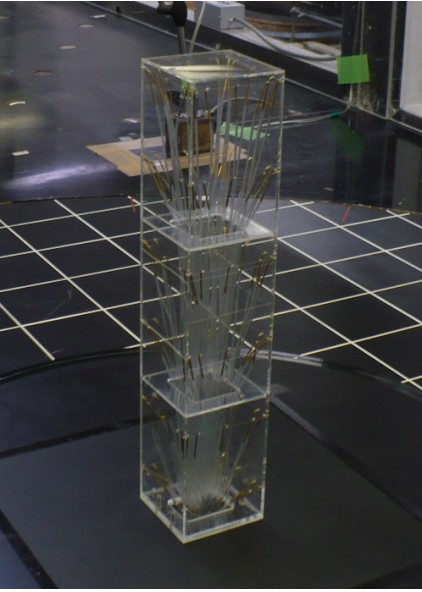

**Figure 4.** Rigid model.

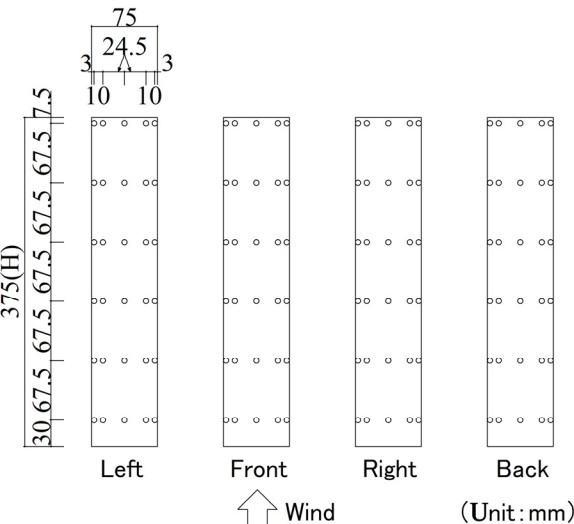

**Figure 5.** Wind pressure measurement points.

A schematic of the wind tunnel experiment using the rigid model is shown in Figure 6. The reference velocity pressure was measured with a pitot-static tube A installed on the windward side of the model. The wind pressure at each measurement point was led from a vinyl tube to the pressure gauge. The pressure on the model surface was measured as the differential pressure from the static pressure obtained by a pitot-static tube B installed at the top of the model. The amplitude and phase distortions of the wind pressure time series data owing to the vinyl tubing were corrected using an appropriate transfer function. The experimental wind speed comprised 15 wind speed steps from approximately 3 to 11 m/s at the reference height of the model, $H$ = 375 mm. Thereafter, the wind speed was expressed as the reduced wind velocity $V_r = U_H/f_0 B$, which is the experimental wind speed at the reference height, $U_H$, divided by the natural frequency of the elastic model, $f_0$ = 8.34 Hz, and the model width, $B$ = 0.075 m. The experimental wind direction was the only direction in which the wind was normal to the front side of the model. The data-sampling frequency for the measurements was 800 Hz. The number of measured data points was 129,465, which corresponds to 90 min in real time, considering the experimental similarity rule for elastic models described later. The distributions of the mean wind pressure coefficients $\overline{C}_{pe}$ and the fluctuating wind pressure coefficient $\widetilde{C}_{pe}$ for smooth and gradient flows for $V_r$ = 9.7 are shown in Figures 7 and 8. The wind pressure coefficients $\overline{C}_{pe}$ and $\widetilde{C}_{pe}$ are defined as follows:

$$\overline{C}_{pe} = \overline{P}/q_H, \tag{1}$$

$$\widetilde{C}_{pe} = \widetilde{P}/q_H. \tag{2}$$

where $t$ is the time, $\overline{P}$ and $\widetilde{P}$ denote the mean and root mean square (RMS) of the wind pressure time series at a measurement point $P(t)$, respectively, $q_H = \rho U_H{}^2/2$ is the velocity pressure, and $\rho$ is the air density.

The distributions of $\overline{C}_{pe}$ and $\widetilde{C}_{pe}$ exhibited different trends depending on the experimental flow. The effect of $V_r$ on the wind pressure coefficient distribution is almost negligible.

*2.3. Experiments Using an Elastic Model*

An overview of the elastic model is shown in Figure 9. The geometry of the model and the wind pressure measurement point locations were similar to those of the rigid model. The elastic model was made of balsa wood, and vinyl tubes were fixed inside the model to prevent vinyl tubes from vibrating. A schematic of the wind tunnel experiments using the elastic model is shown in Figure 10. The elastic model was attached to the vibration test apparatus at the bottom of the turntable and elastically supported by springs. The displacements of the model were measured with laser displacement meters at a position 207.5 mm

downward from the centre of rotation. The measured displacement data were low-pass filtered at 20 Hz to remove the noise components at high frequencies. The displacements were measured in two directions: along-wind and across-wind. The definitions of the axes are shown in Figure 11.

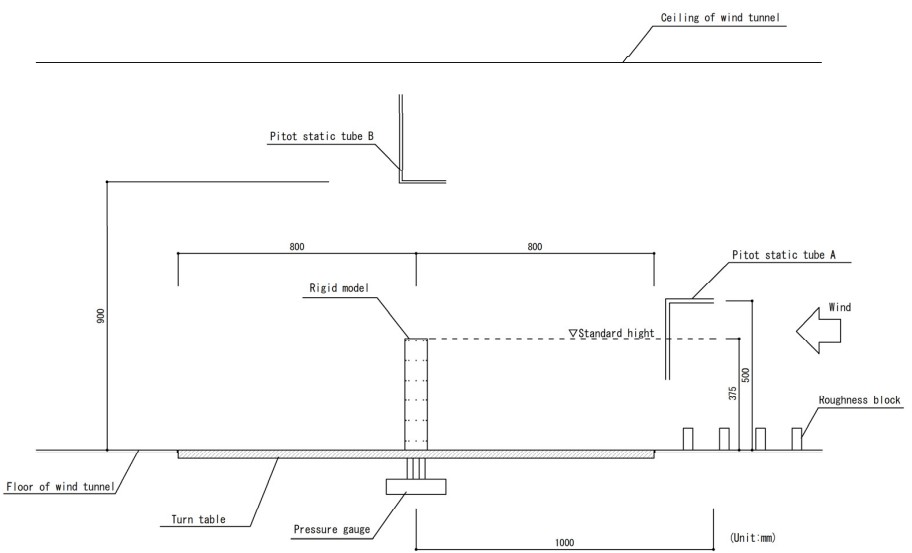

**Figure 6.** Schematic of wind tunnel experiments using the rigid model.

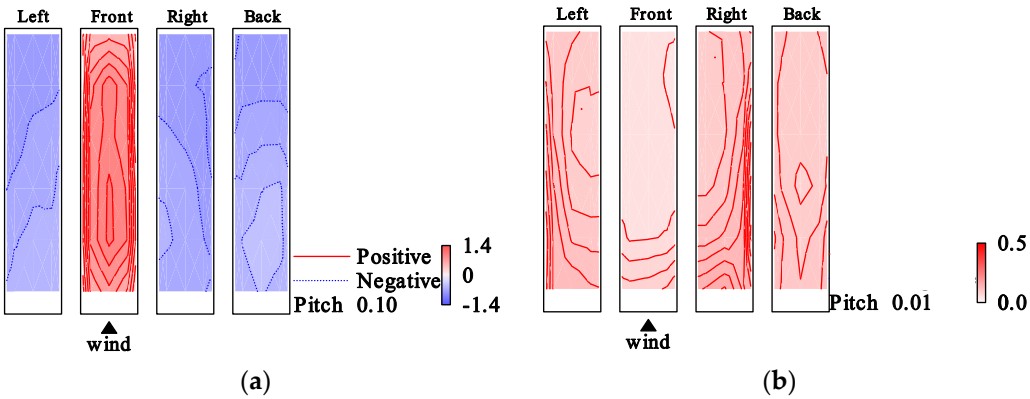

**Figure 7.** Wind pressure coefficient distributions (rigid model in smooth flow, $V_r$ = 9.7): (**a**) $\overline{C}_{pe}$; (**b**) $\widetilde{C}_{pe}$.

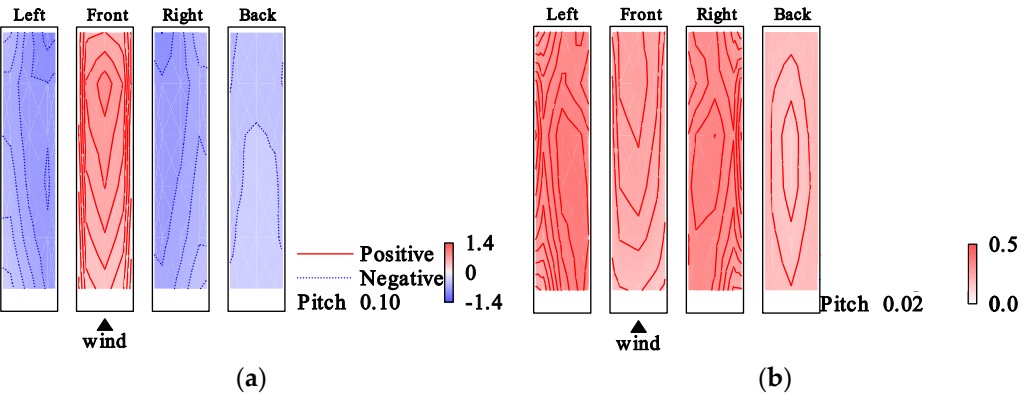

**Figure 8.** Wind pressure coefficient distributions (rigid model in gradient flow, $V_r$ = 9.7): (**a**) $\overline{C}_{pe}$; (**b**) $\widetilde{C}_{pe}$.

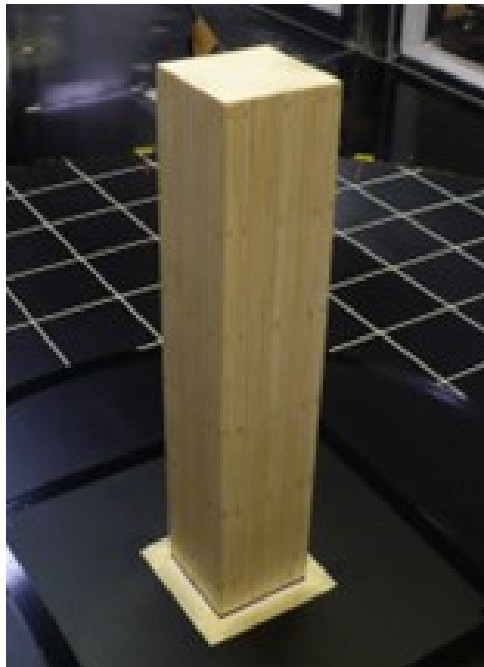

**Figure 9.** Elastic model.

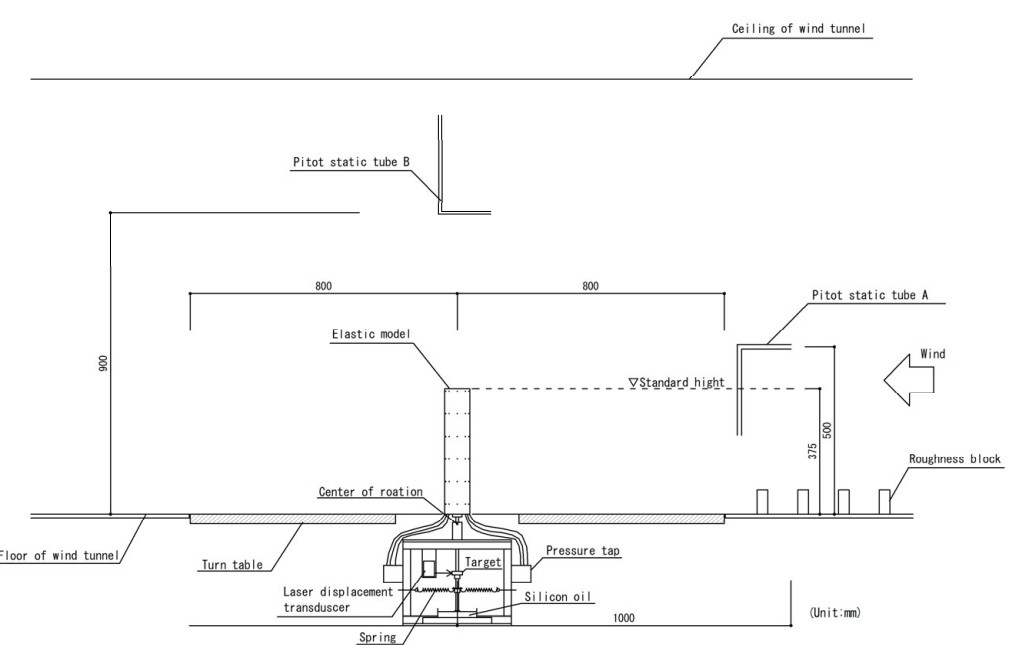

**Figure 10.** Schematic of wind tunnel experiments using the elastic model.

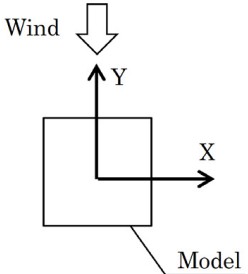

**Figure 11.** Axis definition.

The target of this study was a high-rise RC building that was 30 m wide × 30 m deep × 150 m high, with a building density of 300 kg/m³ and a natural frequency of 0.25 Hz. The damping ratios $h$ were 1% and 2%, respectively. In this experiment, the two-degree-of-freedom rocking vibration was reproduced by matching the reduced wind velocity, mass ratio, and damping ratio between the target building and experimental model according to the similarity rule of the vibration experiment. Table 1 presents the correspondence between the structural properties of the target building and the experimental model. The damping ratios and natural frequencies were evaluated based on the results of the free vibration experiments under no-wind conditions. Free vibration experiments were conducted before and after each experiment in each case. The damping ratios shown in Table 1 were calculated using the following equations by using the average value of the ratio $d$, which is the amplitude of the free vibration waveforms obtained from the free vibration experiments for each adjacent period for approximately 4 s.

$$h = (\ln d/2\pi)/\sqrt{1 + (\ln d/2\pi)^2}. \tag{3}$$

**Table 1.** Corresponding structural specifications between the target building and the experimental model.

| Structural Specification | | | Target Building | Experimental Model |
|---|---|---|---|---|
| Natural frequency (Hz) | | | 0.25 | 8.34 |
| Wind speed at reference height (m/s) | | | 31.0–126.9 | 2.6–10.6 |
| Reduced wind velocity $V_r$ | | | 4.1–16.9 | 4.1–16.9 |
| Building density (kg/m³) | | | 300 | 307 |
| Generalized mass (kg) | | | 13,500,000 | 0.216 |
| Damping ratio $h$ (%) | Smooth flow | Along wind | 1 | 1.08 |
| | | Across wind | | 1.02 |
| | Gradient flow | Along wind | | 1.04 |
| | | Across wind | | 0.96 |
| | Smooth flow | Along wind | 2 | 2.03 |
| | | Across wind | | 1.97 |
| | Gradient flow | Along wind | | 1.97 |
| | | Across wind | | 2.00 |

The relationship between the damping ratios $h_1$, calculated by Equation (3) using the ratio $d$ of amplitudes per period, and the dimensionless displacements $\delta/H$ is shown in Figures 12–15. Here $\delta$ is the model top displacement. In each experiment case, there is almost no amplitude dependency of the damping ratios, and generally stable values are obtained.

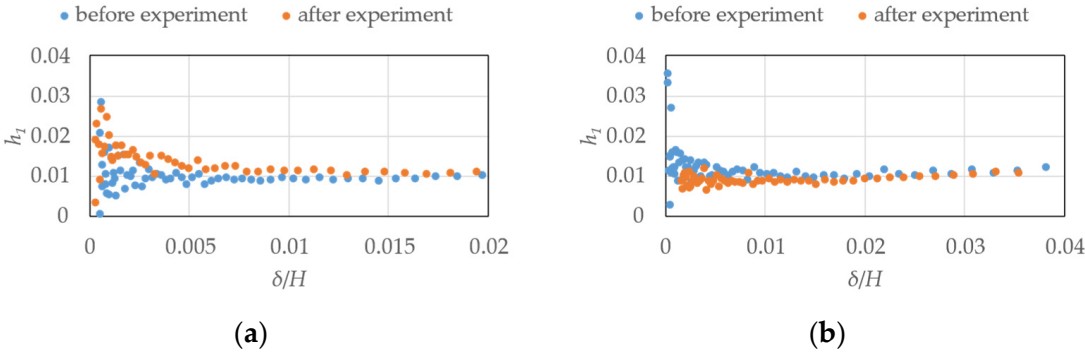

**Figure 12.** Relationship between the displacement ratios and damping ratios of the free vibration experiment (elastic model in smooth flow, $h$ = 1%): (**a**) along wind; (**b**) across wind.

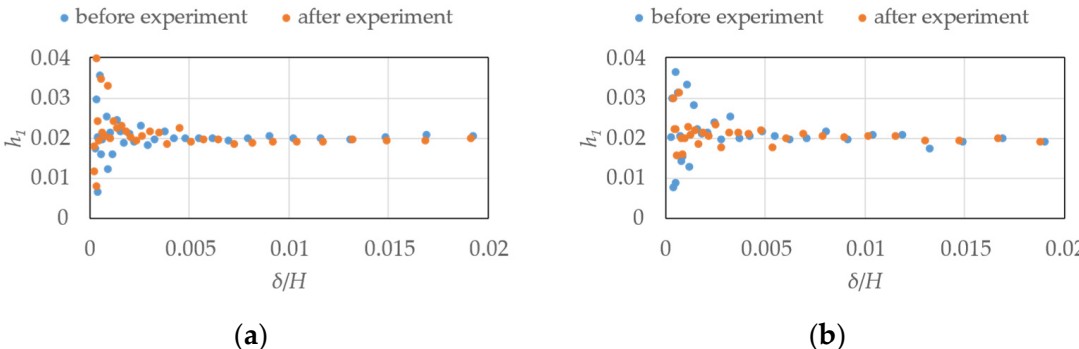

**Figure 13.** Relationship between the displacement ratios and damping ratios of the free vibration experiment (elastic model in smooth flow, *h* = 2%): (**a**) along wind; (**b**) across wind.

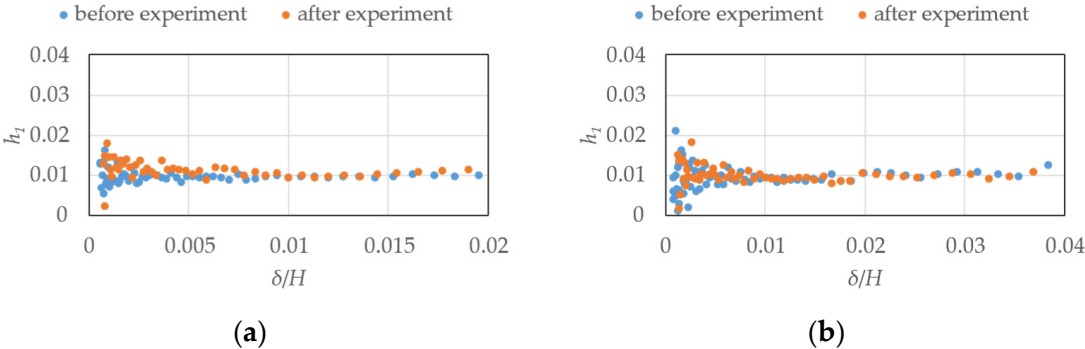

**Figure 14.** Relationship between the displacement ratios and damping ratios of the free vibration experiment (elastic model in gradient flow, *h* = 1%): (**a**) along wind; (**b**) across wind.

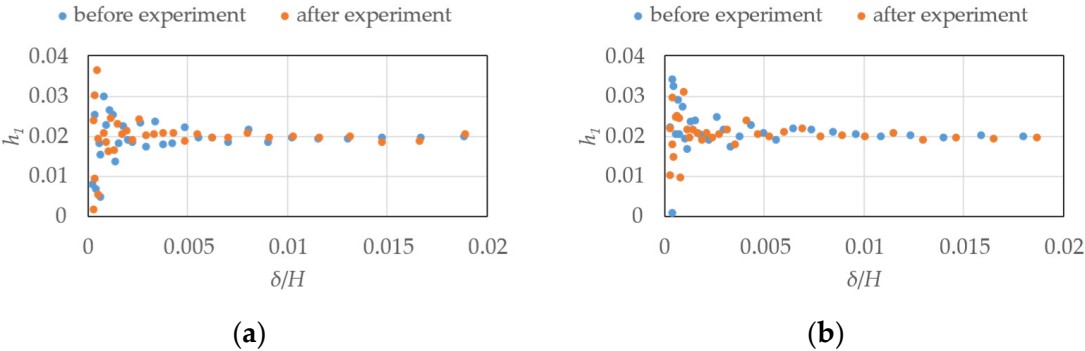

**Figure 15.** Relationship between the displacement ratios and damping ratios of the free vibration experiment (elastic model in gradient flow, *h* = 2%): (**a**) along wind; (**b**) across wind.

The distributions of $\overline{C}_{pe}$ and $\widetilde{C}_{pe}$ for the elastic model in the smooth flow for $h = 1\%$, $h = 2\%$, $V_r = 8.8$, $V_r = 9.7$ are shown in Figures 16–19. For $V_r = 9.7$, $\overline{C}_{pe}$ at the upper part of both side surfaces was smaller than that in the case of $V_r = 8.8$. $\widetilde{C}_{pe}$ shows large values in the leeward region of the model side surfaces compared to those in the case of $V_r = 8.8$ regardless of damping ratio. The different distribution compared to those for the rigid model could be attributed to the large vibration of the elastic model caused by the resonance phenomenon because the reduced wind velocity $V_r = 9.7$ is near the resonance wind speed. The same pattern was observed for the case of $h = 2\%$ in the smooth flow. However, the change in wind pressure distribution was smaller than that for the case of $h = 1\%$. Furthermore, the wind pressure distribution in the two cases of $V_r = 9.7$ and $V_r = 10.7$ was similar to each other for $h = 1\%$, and whereas those for other $V_r$ were almost the same as those at $V_r = 8.8$. For $h = 2\%$, the wind pressure distribution is almost the same

for all other wind speed ranges except for $V_r = 9.7$. The distributions of $\overline{C}_{pe}$ and $\widetilde{C}_{pe}$ for $h = 1\%$, $h = 2\%$, $V_r = 9.7$ in the gradient flow are shown in Figures 20 and 21. In the gradient flow, the wind pressure distributions are almost the same regardless of $h$ or $V_r$.

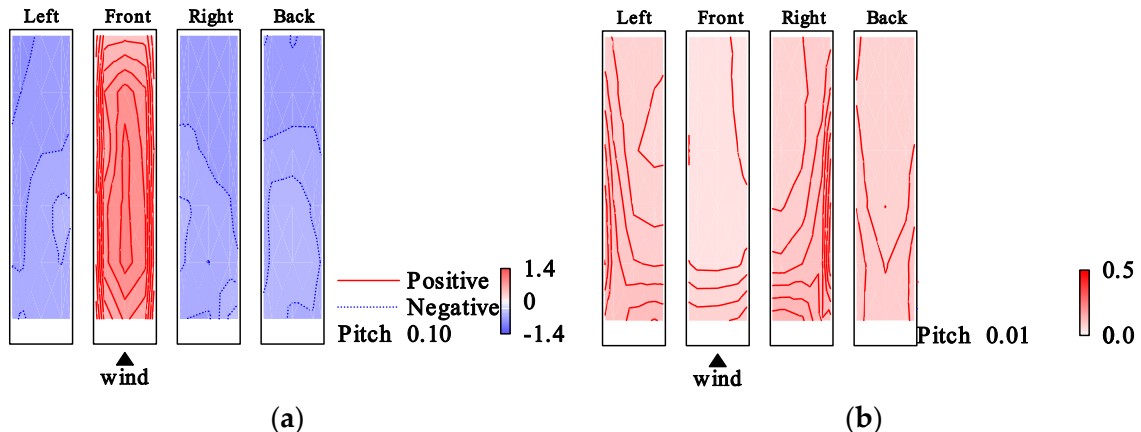

**Figure 16.** Wind pressure coefficient distributions (elastic model in smooth flow, $V_r = 8.8$, $h = 1\%$): (**a**) $\overline{C}_{pe}$; (**b**) $\widetilde{C}_{pe}$.

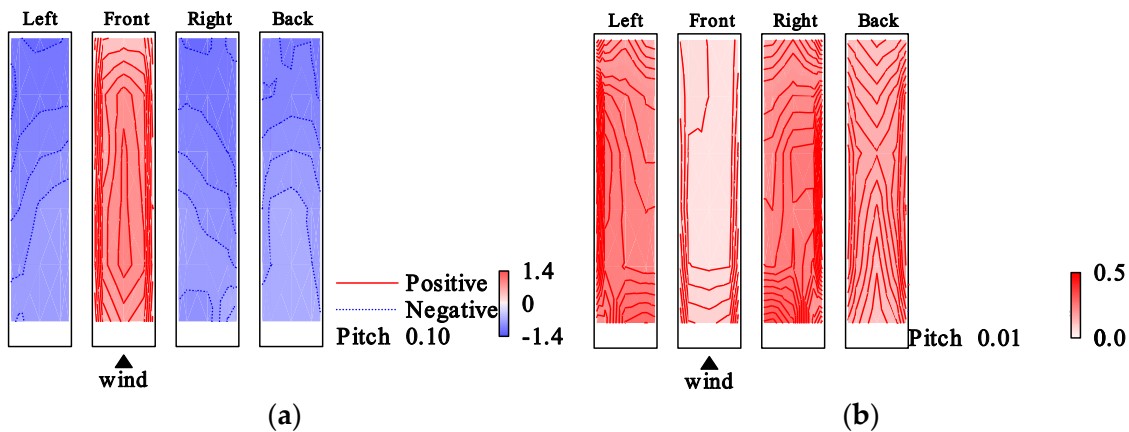

**Figure 17.** Wind pressure coefficient distributions (elastic model in smooth flow, $V_r = 9.7$, $h = 1\%$): (**a**) $\overline{C}_{pe}$; (**b**) $\widetilde{C}_{pe}$.

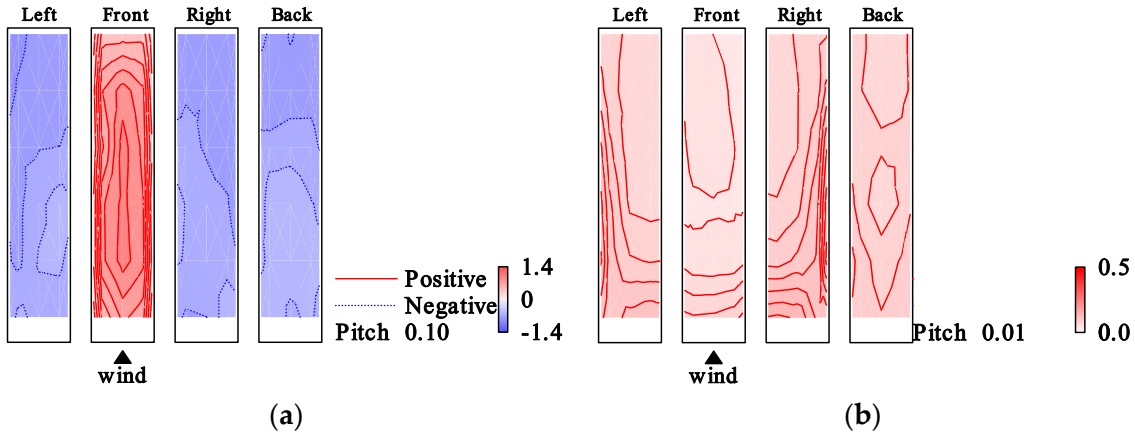

**Figure 18.** Wind pressure coefficient distributions (elastic model in smooth flow, $V_r = 8.8$, $h = 2\%$): (**a**) $\overline{C}_{pe}$; (**b**) $\widetilde{C}_{pe}$.

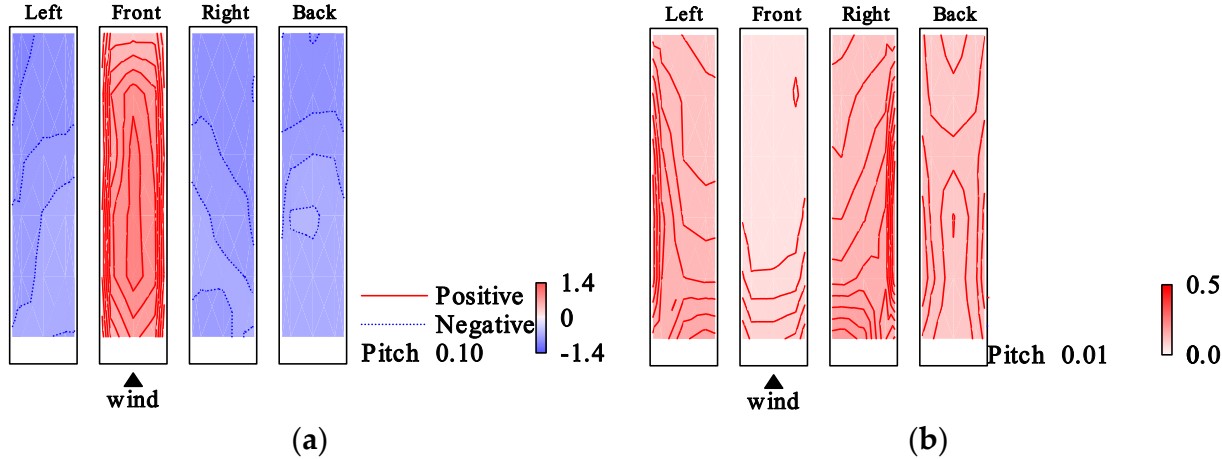

**Figure 19.** Wind pressure coefficient distributions (elastic model in smooth flow, $V_r = 9.7$, $h = 2\%$): (**a**) $\overline{C}_{pe}$; (**b**) $\widetilde{C}_{pe}$.

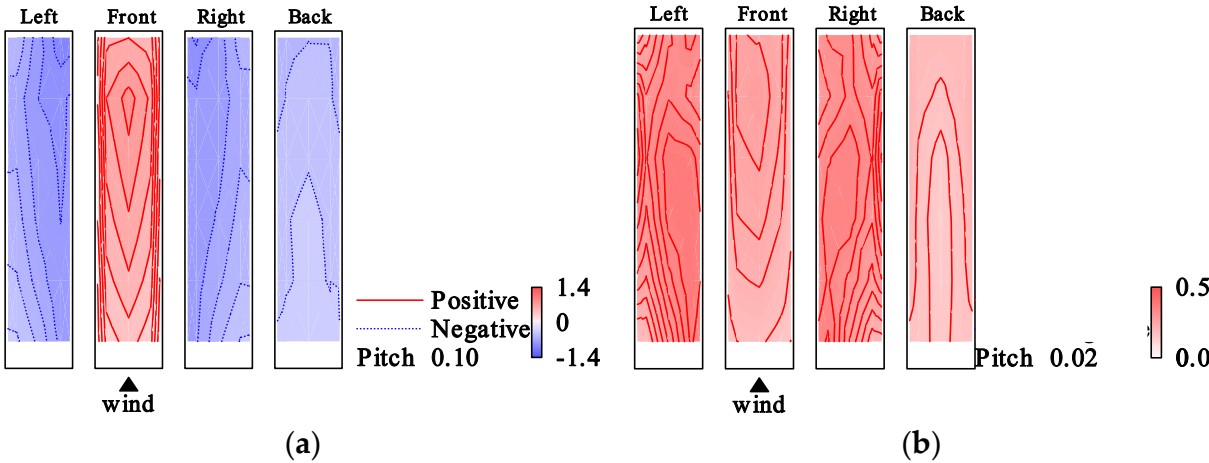

**Figure 20.** Wind pressure coefficient distributions (elastic model in gradient flow, $V_r = 9.6$, $h = 1\%$): (**a**) $\overline{C}_{pe}$; (**b**) $\widetilde{C}_{pe}$.

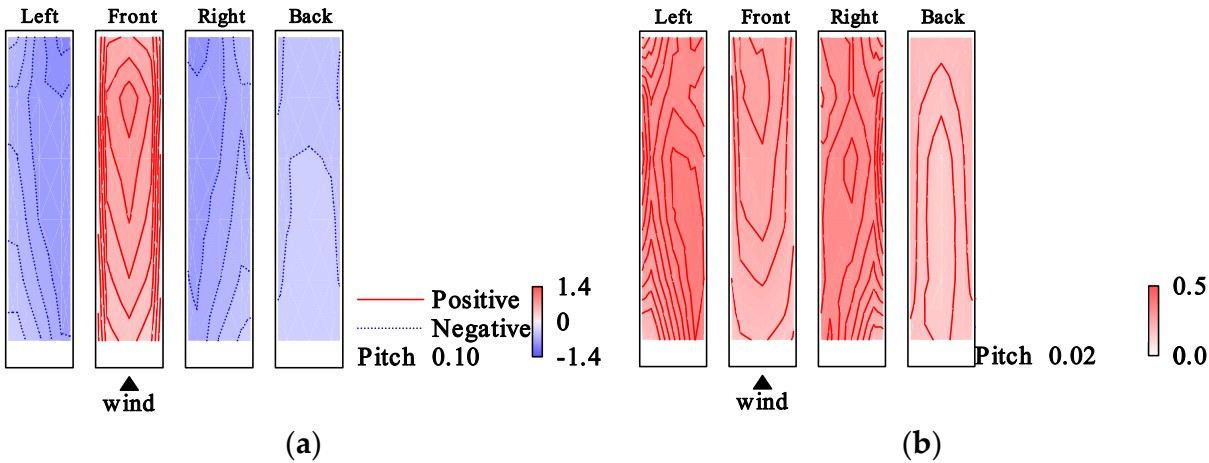

**Figure 21.** Wind pressure coefficient distributions (elastic model in gradient flow, $V_r = 9.6$, $h = 2\%$): (**a**) $\overline{C}_{pe}$; (**b**) $\widetilde{C}_{pe}$.

The generalized wind force $F(t)$ was calculated using $P(z)$ obtained by linear interpolation of the wind force (P1–P6) between each layer, assuming a linear mode $\mu(z)$, as shown in Figure 22, using the following equation:

$$F(t) = \int_{H'}^{H+H'} P(z)\mu(z)dz,$$ (4)

$$\mu(z) = z/(H + H').$$ (5)

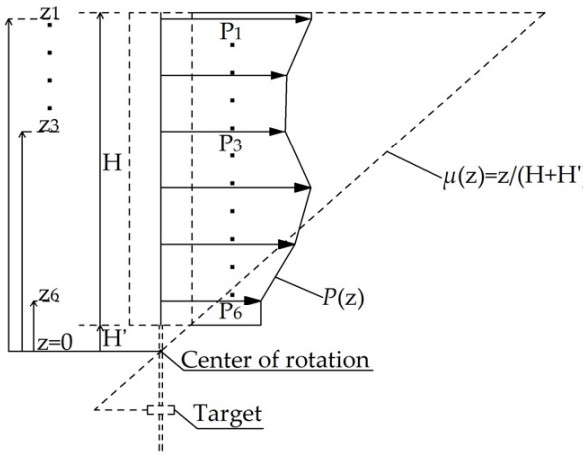

**Figure 22.** Vibration mode.

The time-history response analysis was performed by solving the following equation of motion, using as input the generalized wind force time series $F(t)$ calculated from the wind pressure data for the rigid and elastic models. Newmark's beta method (linear acceleration method) was used for numerical integration.

$$m\ddot{x}(t) + c\dot{x}(t) + kx(t) = F(t).$$ (6)

where $m$ is the generalized mass of 0.216 kg, and $k$ is the generalized stiffness of 592 N/m.

The damping coefficient, $c$ kg/s, was obtained from the damping ratio $h$ for each case presented in Table 1 using the following equation:

$$c = 2h(2\pi f_0)m.$$ (7)

The number of data points for the generalized wind force was set to 143,850, and the first 14,385 data points corresponding to the first 10 min of real time were envelope-processed. In Figures 23–30, the following three displacements are shown for comparison.

$\delta_{\text{me}}$ : displacement of the elastic model by measurement.

$\delta_{\text{ae}}$ : displacement by time-history response analysis using the wind force on the elastic model.

$\delta_{\text{ar}}$ : displacement by time-history response analysis using the wind force on the rigid model.

The mean, RMS, and maximum values of the measured displacements of the elastic model, converted to values at the top of the model, are shown in Figures 23–30 as the ratio to the height of the model. Each statistic is the ensemble average of nine times calculated for 14,385 data which corresponded to a real time of 10 min. The maximum value indicates the maximum absolute displacement. The displacement ratios from the time-response analysis using the generalized wind force $F(t)$ are also shown in Figures 23–30.

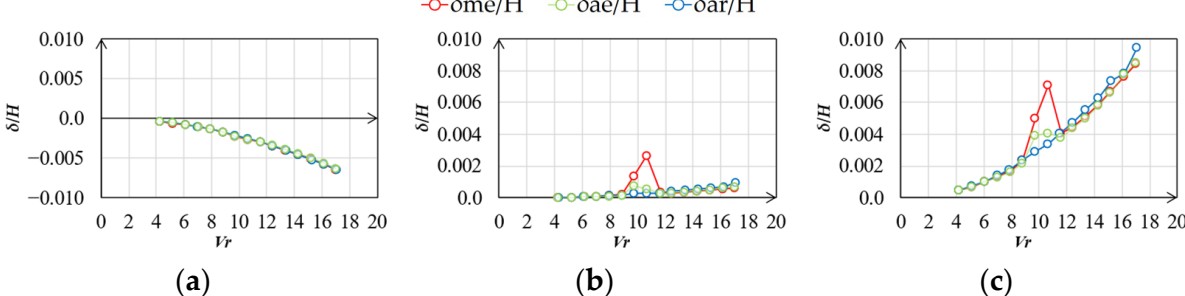

**Figure 23.** Vibration–displacement ratio (smooth flow, *h* = 1%, along wind): (**a**) mean; (**b**) RMS; (**c**) max.

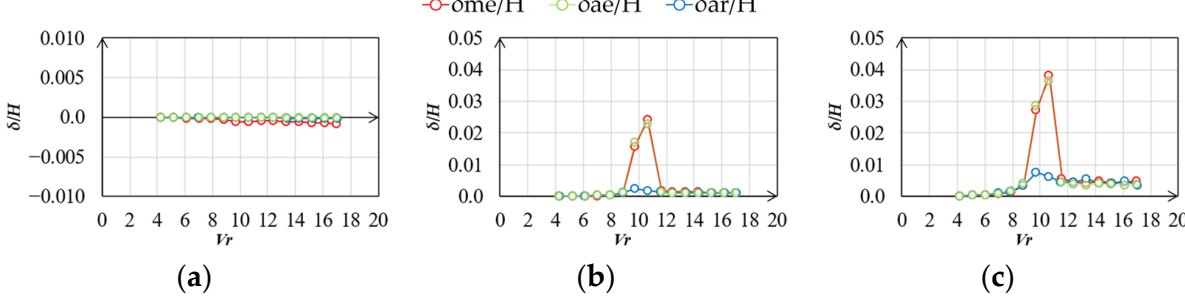

**Figure 24.** Vibration–displacement ratio (smooth flow, *h* = 1%, across wind): (**a**) mean; (**b**) RMS; (**c**) max.

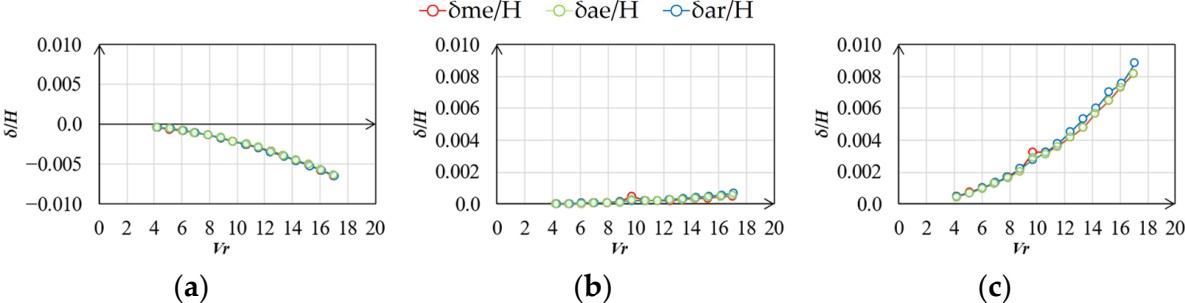

**Figure 25.** Vibration–displacement ratio (smooth flow, *h* = 2%, along wind): (**a**) mean; (**b**) RMS; (**c**) max.

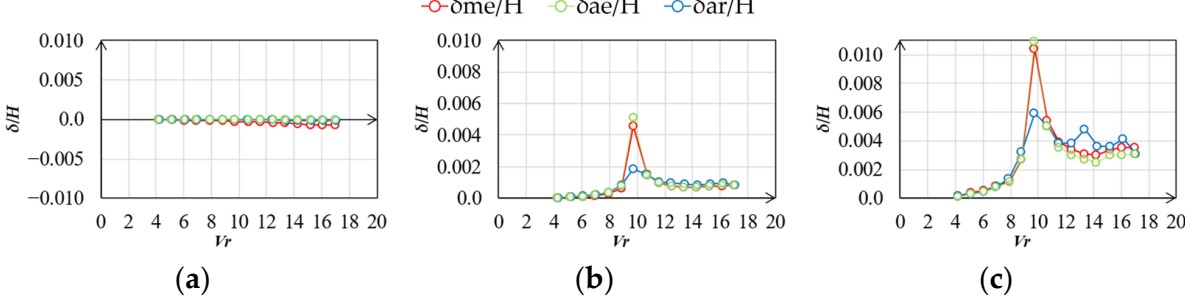

**Figure 26.** Vibration–displacement ratio (smooth flow, *h* = 2%, across wind): (**a**) mean; (**b**) RMS; (**c**) max.

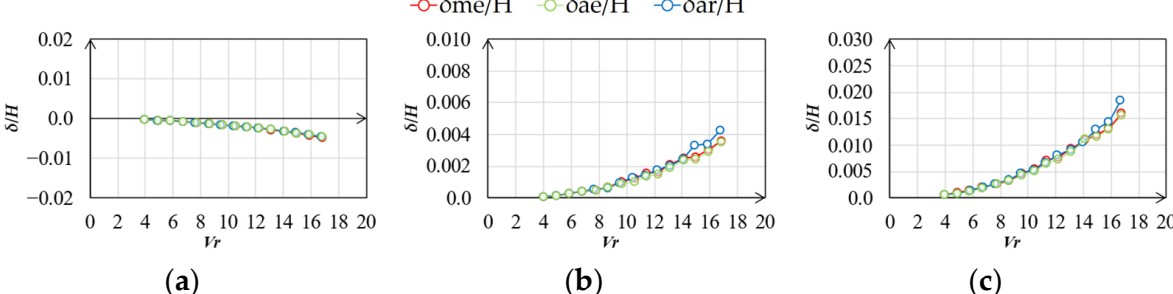

**Figure 27.** Vibration–displacement ratio (gradient flow, $h = 1\%$, along wind): (**a**) mean; (**b**) RMS; (**c**) max.

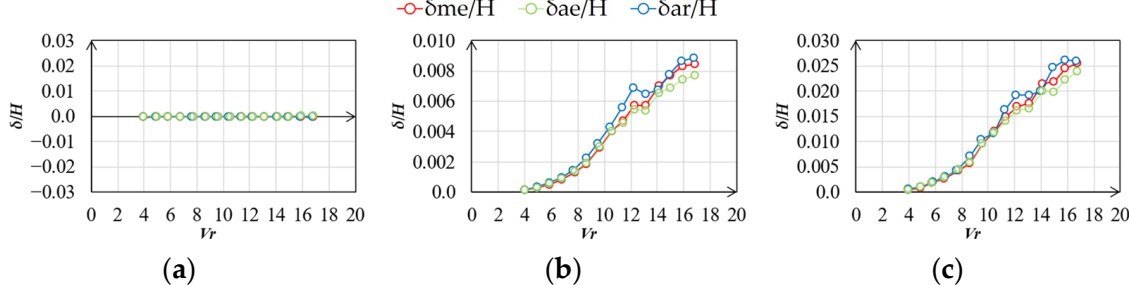

**Figure 28.** Vibration–displacement ratio (gradient flow, $h = 1\%$, across wind): (**a**) mean; (**b**) RMS; (**c**) max.

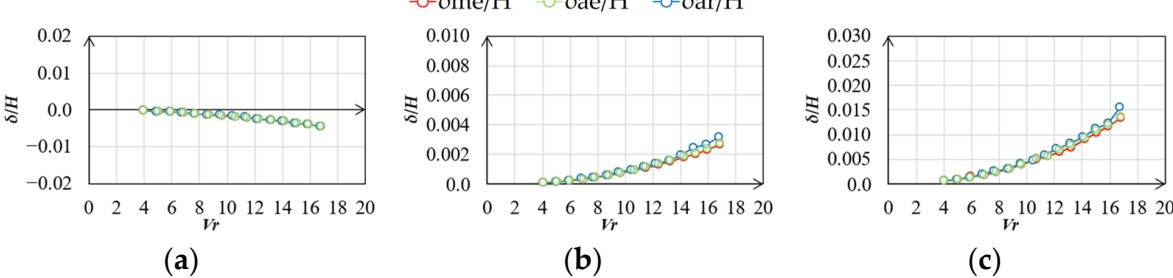

**Figure 29.** Vibration–displacement ratio (gradient flow, $h = 2\%$, along wind): (**a**) mean; (**b**) RMS; (**c**) max.

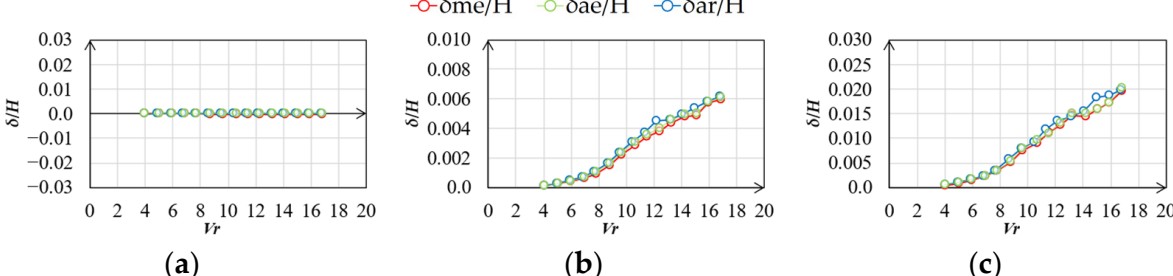

**Figure 30.** Vibration–displacement ratio (gradient flow, $h = 2\%$, across wind): (**a**) mean; (**b**) RMS; (**c**) max.

The statistics of the along- and across-wind vibration displacements in the smooth flow are shown in Figures 23 and 26. The $\delta_{\mathrm{me}}/H$ and $\delta_{\mathrm{ae}}/H$ values are almost the same in the along- and across-wind directions. Meanwhile, the RMS and maximum values of $\delta_{\mathrm{me}}/H$ and $\delta_{\mathrm{ae}}/H$ are larger than those of $\delta_{\mathrm{ar}}/H$ in across-wind direction around $V_{\mathrm{r}} = 9.7$

to 10.7 at $h = 1\%$ and $V_r = 9.7$ at $h = 2\%$. This is attributed to the effect of unsteady wind forces generated by the vibration of the elastic model. $\delta_{me}/H$ was larger than $\delta_{ar}/H$ and $\delta_{ae}/H$ in the same wind speed range along the wind direction. This is because when the model is oscillating significantly across-wind, it oscillates in an elliptical orbit in the XY plane, which also affects the along-wind vibration displacement.

The vibration–displacement ratios of the along- and across-wind directions in the gradient flow are shown in Figures 27–30. There was no significant difference in the along-wind vibration displacement among $\delta_{me}/H$, $\delta_{ae}/H$, and $\delta_{ar}/H$ for the experimental results for the gradient flow. Meanwhile, there was a difference in the RMS and maximum values of the across-wind vibration displacement for $\delta_{ar}/H$, $\delta_{me}/H$, and $\delta_{ae}/H$, although the difference was smaller than that in the case of smooth flow.

## 3. Complex Proper Orthogonal Decomposition

### 3.1. Evaluation Method

The CPOD analysis was performed on the wind pressure data obtained from wind tunnel experiments using rigid and elastic models by solving the following eigenvalue problem:

$$\mathbf{RA\Phi} = \mathbf{\Phi\Lambda}, \tag{8}$$

where $\mathbf{\Phi}$ is a mode matrix with the $j^{th}$ eigenvector as $j$-column; $\mathbf{\Lambda}$ is an eigenvalue matrix with the $j^{th}$ eigenvalues $\lambda_j$ as $j$-row and $j$-column diagonal elements; and $\mathbf{A}$ is a diagonal matrix with a $j$-row and $j$-column element, $A_j$, which is the ratio of the burden area at point $j$ to the total side area. $\mathbf{R}$ is a complex covariance matrix, and assuming that the wind pressure time series at point $j$, $p_j(t)$, is a stationary process with period $T$, the $j$-row and $k$-column elements $R_{jk}$ are obtained as follows:

$$R_{jk} = \frac{1}{T} \int_0^T \widetilde{p}_j(t) \cdot \widetilde{p}_k{}^*(t) dt = \sum_{\omega_n \geq 0} 4 P_j(\omega_n) \cdot P_k{}^*(\omega_n), \tag{9}$$

$$P_j(\omega_n) = \frac{1}{T} \int_0^T p_j(t) \cdot e^{-i\omega_n t} dt, \tag{10}$$

$$\widetilde{p}_j(t) = \sum_{\omega_n \geq 0} 2 P_j(\omega_n) \cdot e^{i\omega_n t}. \tag{11}$$

where $i$ denotes an imaginary unit, $*$ is the complex conjugate, and $P_j(\omega_n)$ is the Fourier coefficient of the wind pressure at point $j$, $p_j(t)$, with respect to frequency $\omega_n$. In this method, the covariance of the fluctuating wind pressure is obtained using the analytical signal $\widetilde{p}_j(t)$ shown in Equation (11), and the eigenmodes and eigenvalues are obtained using a complex eigenvalue analysis for $\mathbf{R}$. Mode $\phi_j$ shall have orthogonality in the broad sense using $\mathbf{A}$, and matrix $\mathbf{\Phi}$ is normalized as follows:

$$\mathbf{\Phi}^\dagger \mathbf{A} \mathbf{\Phi} = \mathbf{I}. \tag{12}$$

where, † is the Hermitian conjugate, and $\mathbf{I}$ is the unit matrix.

### 3.2. Contribution Ratio

From the eigenvalues obtained by the CPOD analysis, the $k^{th}$ contribution ratio $C_k$ was obtained using Equation (13).

$$C_k = \frac{\lambda_k}{\sum_{j=1}^N \lambda_j}. \tag{13}$$

where $N$ is the maximum mode order.

The contribution ratios of the 1st- to 5th-mode, $C_1$ to $C_5$, in smooth and gradient flows for the rigid model are shown in Figure 31. $C_1$ and $C_2$ are remarkably higher than those of the other modes for both smooth and gradient flows. $C_k$ for the elastic model is shown

in Figures 32 and 33. In the smooth flow of $V_r = 9.7$ and $V_r = 10.7$, $C_1$ is remarkably large for $h = 1\%$. In the case of $h = 2\%$, $C_1$ is much larger than the others when $V_r = 9.7$ in the smooth flow. However, the contribution ratios, $C_k$, in the gradient flow are almost the same regardless of the damping ratio, $h$, reduced velocity, $V_r$, and whether the model is rigid or elastic.

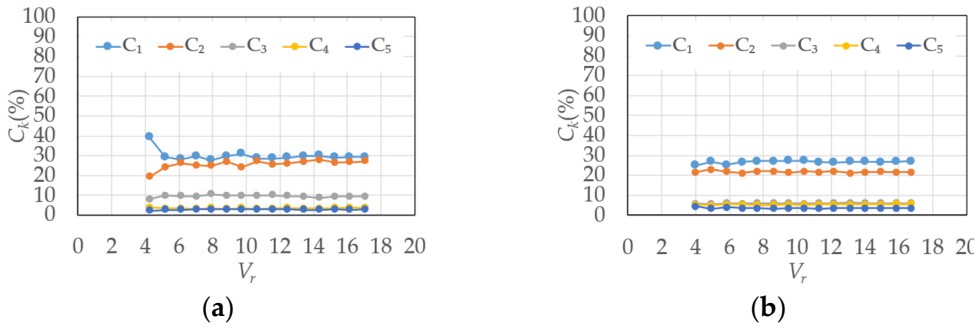

**Figure 31.** Contribution ratio (rigid model): (**a**) smooth flow; (**b**) gradient flow.

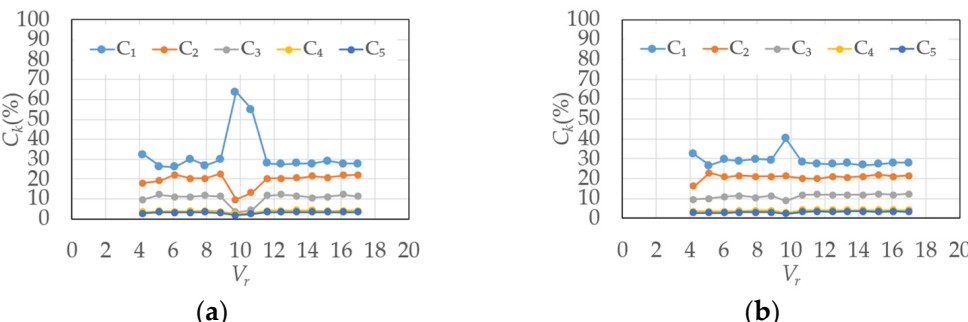

**Figure 32.** Contribution ratio (elastic model in smooth flow): (**a**) $h = 1\%$; (**b**) $h = 2\%$.

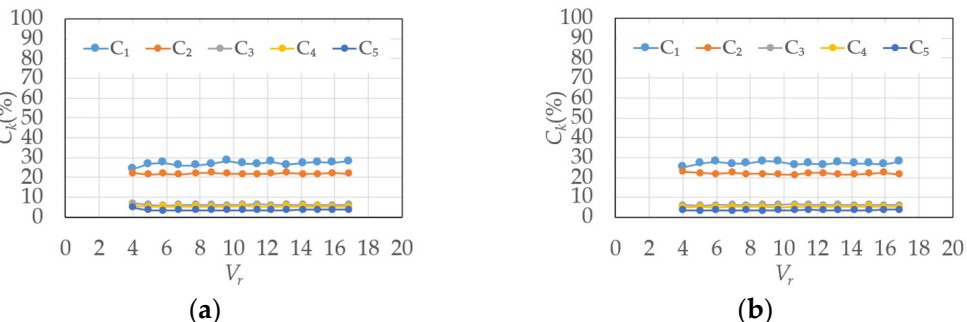

**Figure 33.** Contribution ratio (elastic model in gradient flow): (**a**) $h = 1\%$; (**b**) $h = 2\%$.

### 3.3. Eigenmodes

The eigenmodes obtained by the CPOD analysis are complex and have real and imaginary parts. The absolute value and phase of the eigenmodes, $\left|\phi_{kj}\right|$ and $arg\left(\phi_{kj}\right)$, are important for considering the characteristics of phenomena such as fluctuating wind pressure fields around buildings. $\phi_{kj}$ represents the $j$-row and $k$-column elements of $\mathbf{\Phi}$ and the element at point $k$ of the $j^{\text{th}}$ eigenmode. The 1st- and 2nd-eigenmodes for the rigid model and the elastic model in the smooth flow of reduced velocity $V_r = 9.7$ are shown in Figures 34 and 35, respectively. Regarding the phases on the right and left sides of the rigid model, the 1st-mode is the symmetric mode, whereas the 2nd-mode is the anti-symmetric mode. Meanwhile, the eigenmodes of the elastic model shown in Figure 35 are opposite to those of the rigid model, with the 1st-mode being the anti-symmetric mode and the 2nd-mode being the symmetric mode. The eigenmodes for the rigid model and the elastic

model in the gradient flow of reduced velocity $V_r = 9.6$ are shown in Figures 36 and 37, respectively. The 1st-mode is the symmetric mode and the 2nd-mode is the anti-symmetric mode, with no significant difference in absolute value and phase for both the rigid and elastic models. However, the absolute value distributions and phase characteristics are different from those in the smooth flow.

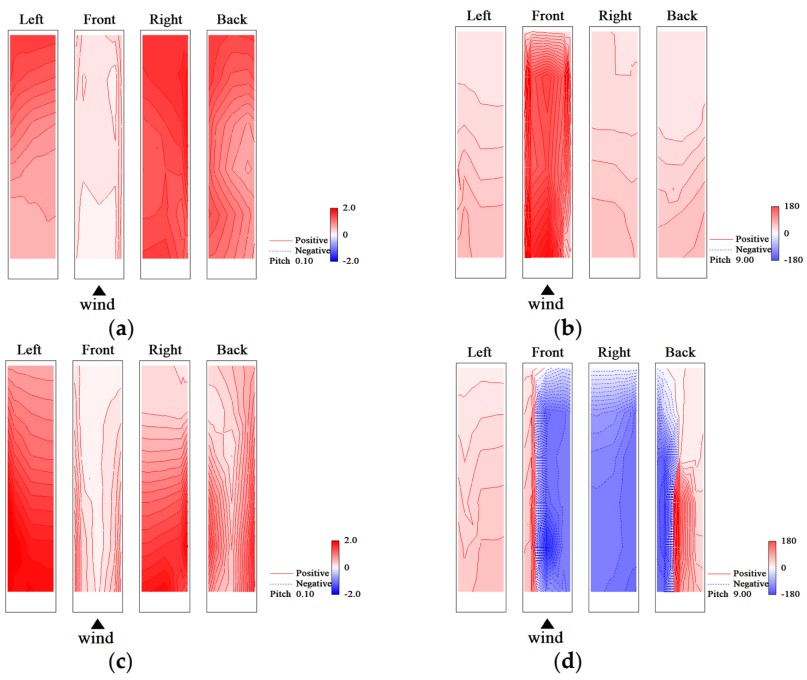

**Figure 34.** Eigenmodes (rigid model in smooth flow, $V_r = 9.7$): (**a**) 1st-mode, absolute value; (**b**) 1st-mode, phase; (**c**) 2nd-mode, absolute value; (**d**) 2nd-mode, phase.

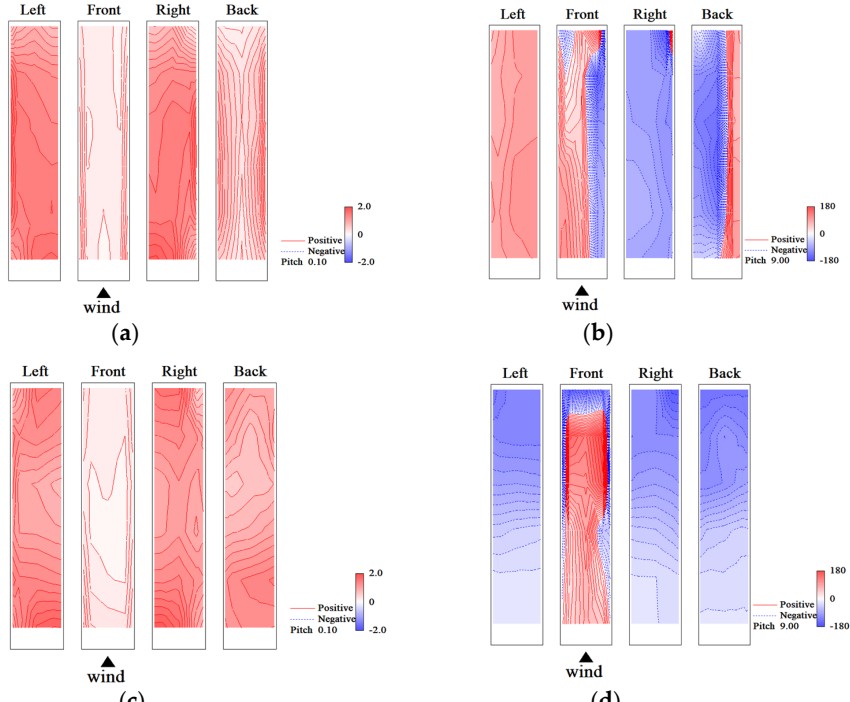

**Figure 35.** Eigenmodes (elastic model in smooth flow, $V_r = 9.7$, $h = 1\%$): (**a**) 1st-mode, absolute value; (**b**) 1st-mode, phase; (**c**) 2nd-mode, absolute value; (**d**) 2nd-mode, phase.

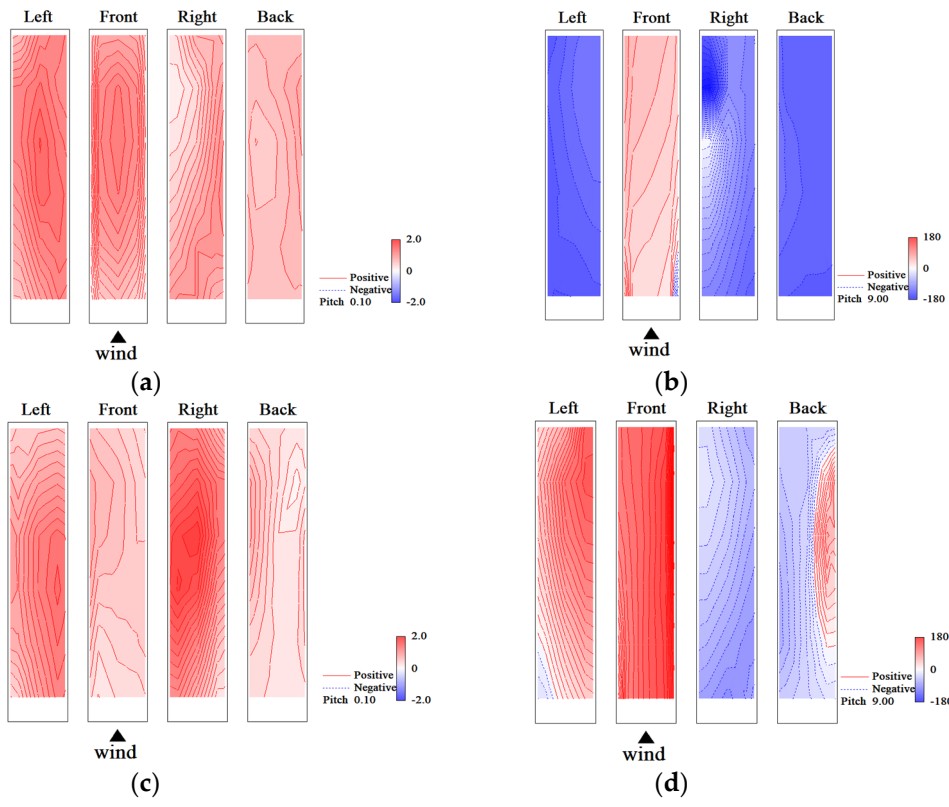

**Figure 36.** Eigenmodes (rigid model in gradient flow, $V_r = 9.6$): (**a**) 1st-mode, absolute value; (**b**) 1st-mode, phase; (**c**) 2nd-mode, absolute value; (**d**) 2nd-mode, phase.

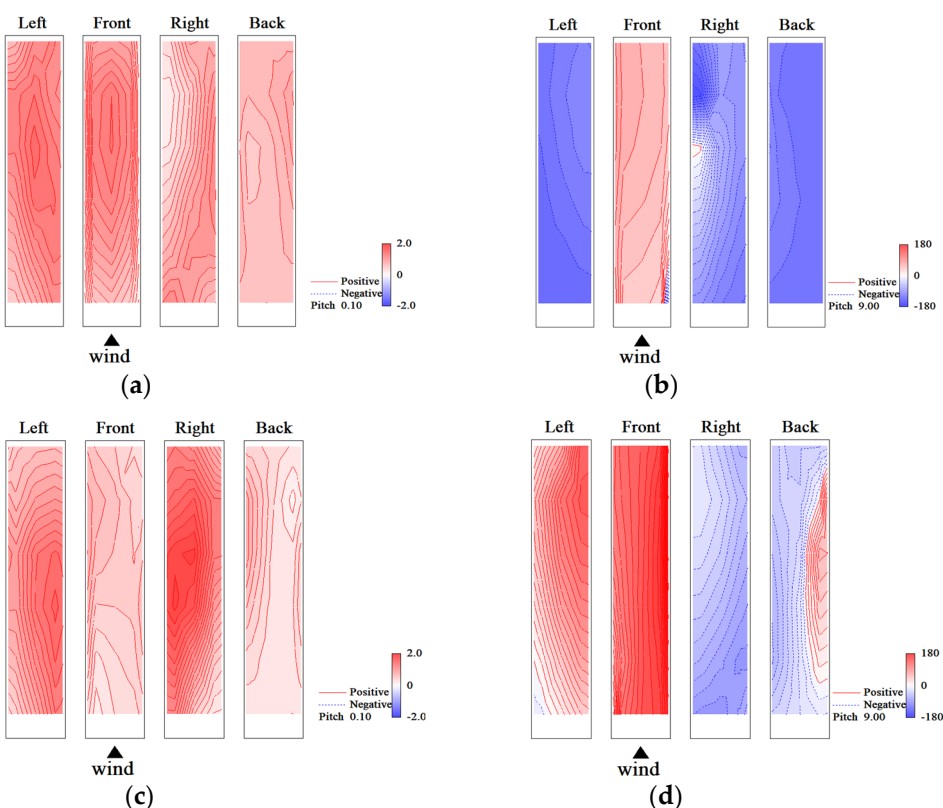

**Figure 37.** Eigenmodes (elastic model in gradient flow, $V_r = 9.6$, $h = 1\%$): (**a**) 1st-mode, absolute value; (**b**) 1st-mode, phase; (**c**) 2nd-mode, absolute value; (**d**) 2nd-mode, phase.

## 4. Evaluation of Phase Characteristics

*4.1. Symmetricity of Fluctuating Wind Pressure Mode*

This section proposes a method for quantitatively evaluating the symmetricity of the fluctuating wind pressure modes of CPOD.

### 4.1.1. Inner Product

The vectors $a$ and $b$ are assumed in Equations (14) and (15).

$$a = \{a_1, \cdots, a_j, \cdots, a_N\}^T, \tag{14}$$

$$b = \{b_1, \cdots, b_j, \cdots, b_N\}^T. \tag{15}$$

The inner product $a \cdot b$ of vectors $a$ and $b$ is defined as follows:

$$a \cdot b = \sum_{j=1}^{N} a_j^* A_j b_j. \tag{16}$$

where $A_j$ is the barden area of point $j$.

The magnitudes of $a$, $|a|$, whose elements are complex, are defined as follows:

$$|a| = \sqrt{a \cdot a} = \sqrt{\sum_{j=1}^{n} a_j^* A_j a_j}. \tag{17}$$

### 4.1.2. Element Exchange Vector

The element exchange vector of $a$ is defined as Equation (18):

$$a_e = \left\{ a_{1'}, \cdots, a_{j'}, \cdots, a_{N'} \right\}^T \tag{18}$$

where $j'$ denotes the number of points symmetrical to point $j$ with respect to the central section ($X = 0$ in Figure 11). Because the pressure measuring points are located symmetrically with respect to the central section, the weights $A_j$ and $A_{j'}$ are equal, $A_j = A_j$. In the case where point $j$ is on the centerline of the front side, $j' = j$. According to Equations (14)–(18), the inner product $a \cdot a_e$ and $|a_e|$ can be expressed as follows:

$$a \cdot a_e = \sum_{j=1}^{N} a_j^* A_j a_{j'}, \tag{19}$$

$$|a_e| = \sqrt{\sum_{j=1}^{N} a_{j'}^* A_{j'} a_{j'}}. \tag{20}$$

Because the complex conjugate of $a_j^* A_j a_{j'}$ $a_j^* A_j a_j$, are always included in Equation (19), $a \cdot a_e$ is always a real number. Considering Equations (17) and (20), all of $a_{j'}^* A_{j'} a_{j'}$ are included in Equation (17) and vice versa, so the following equation holds:

$$|a| = |a_e|. \tag{21}$$

### 4.1.3. Symmetry Index

The symmetry index of vector $a$, $I_s$, is defined in Equation (22):

$$I_s(a) = \frac{a \cdot a_e}{|a||a_e|}. \tag{22}$$

Here, it is an example of a symmetry index $I_s$ using eigenmodes consisting of three elements with the same burden area. If points 1 and 3 are symmetrically located and $2' = 2$, the element exchanged vector of the vector $\boldsymbol{a} = \{a_1, a_2, a_3\}^T$ is the vector $\boldsymbol{a}_e = \{a_{1'}, a_{2'}, a_{3'}\}^T = \{a_3, a_2, a_1\}^T$. When $a_1 = a_3$, the symmetry index $I_s(\boldsymbol{a}) = 1$, and when $a_1 = -a_3$ and $a_2 = 0$, $I_s(\boldsymbol{a}) = -1$. Thus, the symmetry index $I_s(\boldsymbol{a})$ expresses the symmetry of vector $\boldsymbol{a}$.

4.1.4. Evaluation Results

The relationship between $I_s$ and $V_r$ for a rigid model in the smooth flow is shown in Figure 38. $I_s$ for the rigid model varies significantly between $-1$ and 1 based on the changes in $V_r$. Meanwhile, the symmetry indices for the elastic model for $h = 1\%$, shown in Figure 39, show a clear pattern where $I_s(\boldsymbol{\phi}_1) = -1$ and $I_s(\boldsymbol{\phi}_2) = 1$ in the resonant region of $V_r = 9.7$ or 10.7. Therefore, the 1st-mode is anti-symmetric, the 2nd-mode is symmetric, and the anti-symmetric mode becomes dominant, considering the contribution ratio in Figure 32. In the case of $h = 2\%$ in Figure 40, this pattern is limited to the reduced velocity $V_r = 9.7$. However, the opposite trend is observed in the other velocity regions.

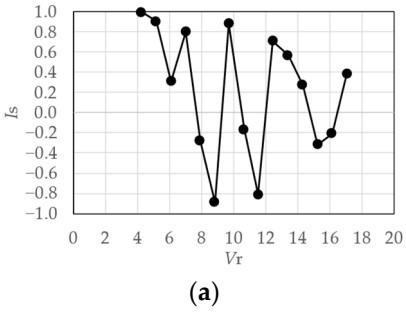 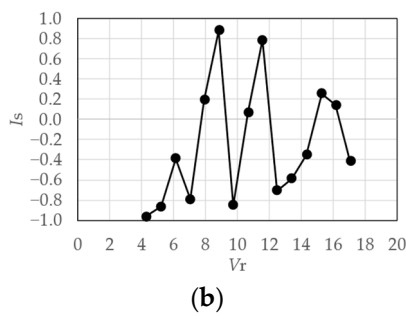

(**a**) (**b**)

**Figure 38.** Symmetry index of eigenmodes (rigid model in smooth flow): (**a**) 1st-mode; (**b**) 2nd-mode.

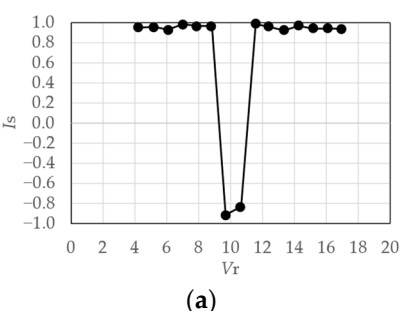 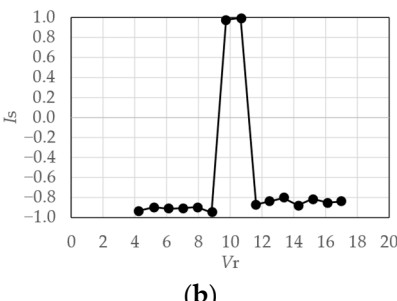

(**a**) (**b**)

**Figure 39.** Symmetry index of eigenmodes (elastic model in smooth flow, $h = 1\%$): (**a**) 1st-mode; (**b**) 2nd-mode.

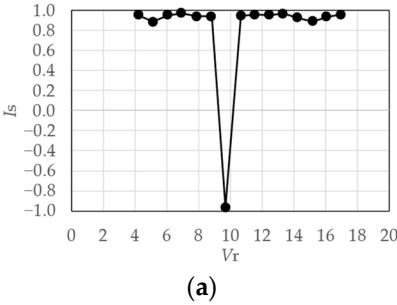 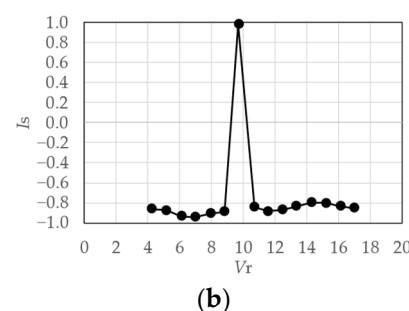

(**a**) (**b**)

**Figure 40.** Symmetry index of eigenmodes (elastic model in smooth flow, $h = 2\%$): (**a**) 1st-mode; (**b**) 2nd-mode.

Figure 41 shows the relationship between $I_s$ and $V_r$ for the rigid model in the gradient flow. The symmetry indexes of the 1st-mode are relatively large, despite the pattern being

less explicit compared to the cases of smooth flow. The anti-symmetricities of the 2nd-mode are weaker than those in the case of smooth flow. The trends in the case of the elastic model shown in Figures 42 and 43 are almost identical to those in the case of the rigid model.

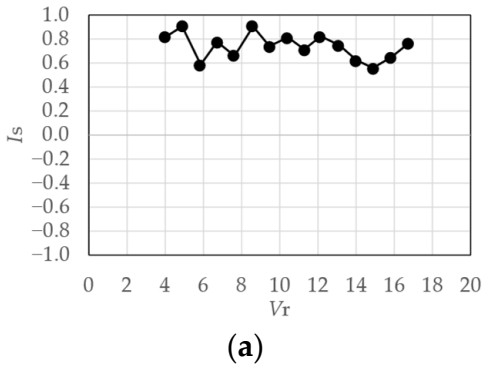 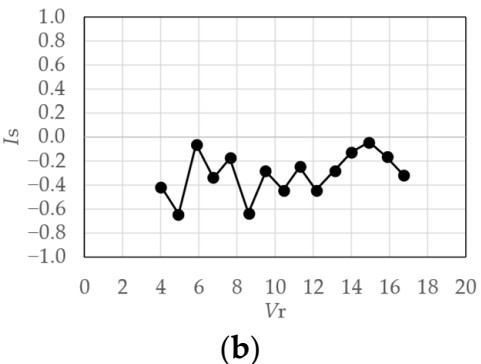

**(a)** **(b)**

**Figure 41.** Symmetry index of eigenmodes (rigid model in gradient flow): (**a**) 1st-mode; (**b**) 2nd-mode.

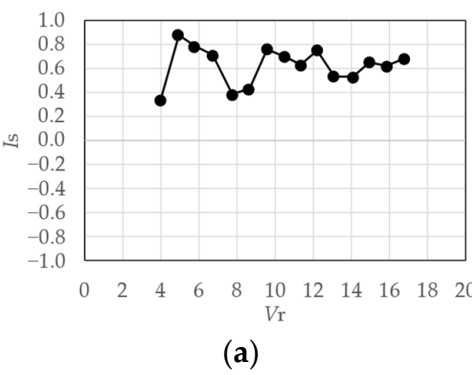 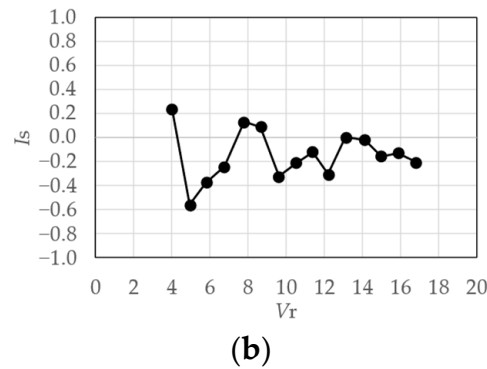

**(a)** **(b)**

**Figure 42.** Symmetry index of eigenmodes (elastic model in gradient flow, $h = 1\%$): (**a**) 1st-mode; (**b**) 2nd-mode.

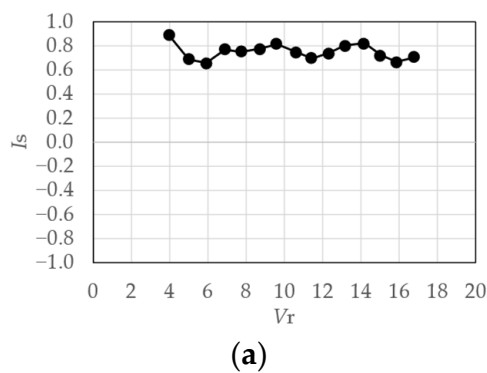 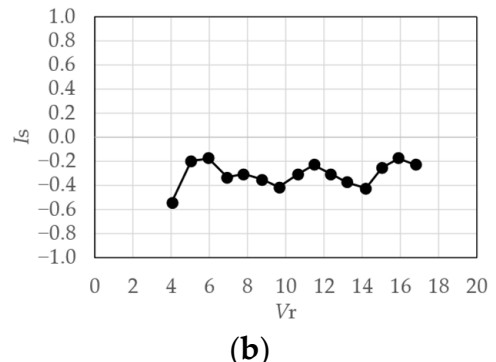

**(a)** **(b)**

**Figure 43.** Symmetry index of eigenmodes (elastic model in gradient flow, $h = 2\%$): (**a**) 1st-mode; (**b**) 2nd-mode.

### 4.2. Similarity Rate of Fluctuating Wind Pressure Fields

As mentioned in the previous section, the symmetry index of the 1st- and 2nd-modes of the fluctuating wind pressure varied with the experimental parameters. This section proposes a method to evaluate the similarity of fluctuating wind pressure fields as the projection ratio of the planes formed by the 1st- and 2nd-eigenmodes, $a_1$ and $a_2$, to that of the other two eigenmodes, $e_1$ and $e_2$. Where vectors $a_1$ and $a_2$, $e_1$ and $e_2$ are normalized, the projection of mode $a_1$ onto the plane formed by modes $e_1$ and $e_2$, is expressed as

$$\mathbf{proj}(a_1) = (e_1 \cdot a_1)e_1 + (e_2 \cdot a_1)e_2. \tag{23}$$

The projection of mode $a_2$ onto that plane, $\mathbf{proj}(a_2)$, can also be obtained, and the following relationship is obtained:

$$\left\{\begin{matrix} \mathbf{proj}(a_1) \\ \mathbf{proj}(a_2) \end{matrix}\right\} = \begin{bmatrix} e_1 \cdot a_1 & e_2 \cdot a_1 \\ e_1 \cdot a_2 & e_2 \cdot a_2 \end{bmatrix} \left\{\begin{matrix} a_1 \\ a_2 \end{matrix}\right\}. \tag{24}$$

The $2 \times 2$ matrix on the right side of Equation (24) is considered the Jacobi matrix, whose determinant represents the projection ratio between the plane formed by $a_1$ and $a_2$ and that formed by $e_1$ and $e_2$. Subsequently, the Jacobian, which represents the magnification rate of the transformation, can be expressed by the following equation:

$$\left| det \begin{bmatrix} (e_1 \cdot a_1) & (e_2 \cdot a_1) \\ (e_1 \cdot a_2) & (e_2 \cdot a_2) \end{bmatrix} \right| = |(e_1 \cdot a_1)(e_2 \cdot a_2) - (e_2 \cdot a_1)(e_1 \cdot a_2)|. \tag{25}$$

The projection rate expresses the similarity of the fluctuating pressure fields consisting of the 1st- and 2nd-eigenmodes. The results of the quantitative evaluation of the similarity between the fluctuating wind pressure modes are shown in Figure 44. The principal axes of the 1st- and 2nd-eigenvectors, $e_1$ and $e_2$, are the eigenmodes of the measurement cases of (a) and (b) shown in Figure 44. In the smooth flow for $h = 1\%$ and $h = 2\%$, the congruency of the plane formed by the two principal eigenmodes by CPOD was different in the resonant wind speed range, indicating different fluctuating wind pressure fields compared to those in other cases. Notably, the damping ratios may affect the fluctuating wind pressure fields by altering the vibration amplitude. Furthermore, an insignificant difference was noted in the characteristics of fluctuating wind pressure fields in the gradient flow. The previous section showed that the symmetricity of fluctuating wind pressure modes tends to vary with the measurement conditions and wind speed. However, by evaluating the similarities of the pressure fields consisting of the 1st- and 2nd-eigenvectors, it appears that the fluctuating wind pressure fields can be classified into the three categories presented in Table 2.

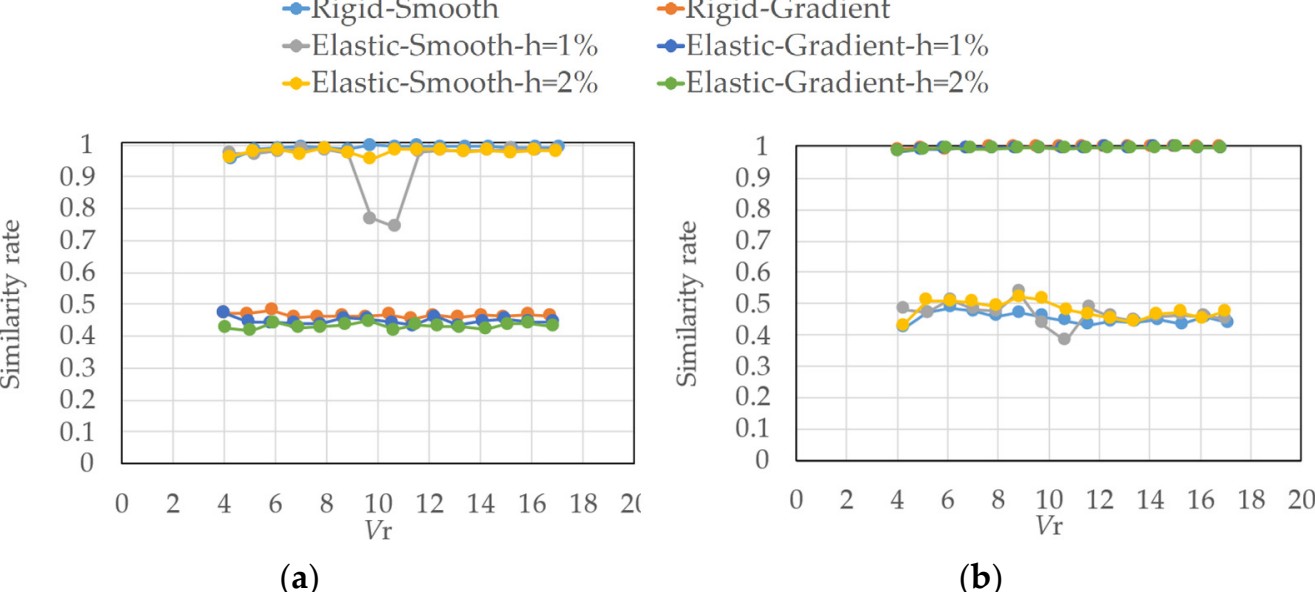

**Figure 44.** Similarity of fluctuating wind pressure fields consisting of two principal eigenmodes: (**a**) similarity to the rigid model case of $V_r = 9.7$ in smooth flow; (**b**) similarity to the rigid model case of $V_r = 9.5$ in gradient flow.

**Table 2.** Classification of fluctuating wind pressure fields.

| Type of Fluctuating Wind Pressure Field | Model | Flow | $h$ | $V_r$ |
|---|---|---|---|---|
| Type 1 (Smooth flow without resonance) | Rigid | Smooth | - | All measured $V_r$ |
| | Elastic | | 1% | Except 9.7 and 10.7 |
| | | | 2% | All measured $V_r$ |
| Type 2 (Smooth flow with resonance) | Elastic | Smooth | 1% | 9.7, 10.7 |
| Type 3 (Gradient flow) | Rigid | Gradient | - | All measured $V_r$ |
| | Elastic | | 1% | |
| | | | 2% | |

## 5. Recomposition of Fluctuating Wind Pressure Fields

In this section, the characteristics of the fluctuating wind pressure fields for the three types in Table 2 are discussed using the recomposition of the principal modes.

### 5.1. Principal Coordinate

A wind pressure vector $\widetilde{p} = \{\widetilde{p}_1(t), \cdots, \widetilde{p}_N(t)\}^T$ with the analytical signal, $\widetilde{p}_j(t)$, of the wind pressure time series at measurement point $j$, $p_j(t)$, as the $j^{\text{th}}$-row element is expressed as the product of CPOD mode matrix $\mathbf{\Phi}$ and the principal coordinate vector $\widetilde{a} = \{\widetilde{a}_1(t), \cdots, \widetilde{a}_N(t)\}^T$ as follows:

$$\widetilde{p} = \mathbf{\Phi}\widetilde{a}. \tag{26}$$

Assuming the normality of eigenmodes in Equation (12), the following relationship is obtained from Equation (26):

$$\mathbf{\Phi}^\dagger \mathbf{A}\widetilde{p} = \widetilde{a}. \tag{27}$$

Therefore, the $j^{\text{th}}$-principal coordinate, $\widetilde{a}_j(t)$, is:

$$\widetilde{a}_j(t) = \boldsymbol{\phi}_j^\dagger \mathbf{A}\widetilde{p}. \tag{28}$$

The PSDs of the principal coordinates are shown in Figure 45. The power spectrum $s(f)$ is nondimensionalized by the model width $B$, mean wind speed at the reference height $U_H$, and variance of the principal coordinates $\sigma^2$.

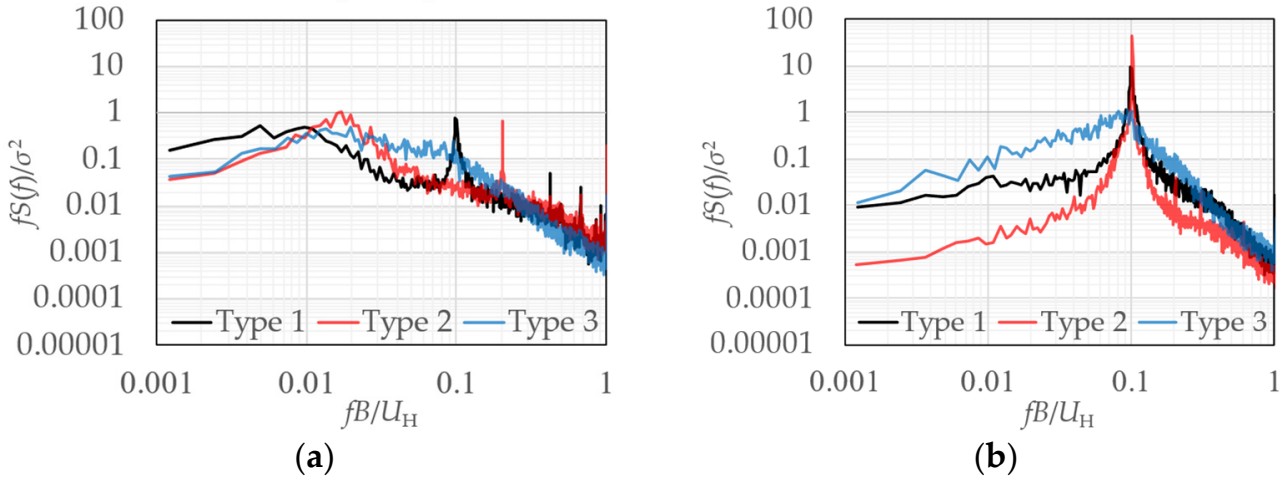

**Figure 45.** Power spectral densities (PSDs) of the principal coordinate: (**a**) symmetric mode; (**b**) anti-symmetric mode.

The PSDs have relatively high power in the low-frequency region for all types in the symmetric mode. The PSD of Type 1 has a sharp peak around $fB/U_H = 0.1$ because the symmetry index of the symmetry mode of Type 1 has a relatively small value of 0.88.

In the case of the anti-symmetric mode, the PSDs of Type 1 and Type 2 have a sharp peak around $fB/U_H = 0.1$ because of the Kalman vortex shedding. In the case of Type 2, the natural frequencies of the elastic model and vortex shedding frequencies are similar; therefore, the resonance phenomenon affects the wind pressure fields. In the case of Type 3, there is a gradual peak at a frequency of approximately 0.1. The peak is not as sharp as that of Type 1 and Type 2 because the vortices are not cleanly generated owing to the high turbulence intensity of the flow.

### 5.2. Correlation of Wind Pressure Field

The correlation coefficient of the eigenmodes is expressed by the following equation:

$$r_{sym,jk} = \left| \Phi_{sym,Typej} \cdot \Phi_{sym,Typek} \right|, \tag{29}$$

$$r_{anti,jk} = \left| \Phi_{anti,Typej} \cdot \Phi_{anti,Typek} \right|. \tag{30}$$

where $\Phi_{sym,Typej}$ and $\Phi_{anti,Typej}$ are the eigenvectors of the symmetric and anti-symmetric modes in the case of Type $j$, respectively.

The correlation coefficients between Types $j$ and $k$, $r_{sym,jk}$, $r_{anti,jk}$, are listed in Table 3. In the case of symmetric modes, the fluctuating wind pressure field in the smooth flow without resonant phenomena, Type 1, is relatively high correlated to that of Types 2 and 3. In the case of anti-symmetric modes, the fluctuating wind pressure fields in the case of smooth flow tend to be close, regardless of resonant phenomena, although the wind pressure field in the gradient flow is different from that in the smooth flow.

**Table 3.** Correlation of fluctuating wind pressure modes.

| Type of Fluctuating Wind Pressure Field | | 1 | 2 | 3 | |
|---|---|---|---|---|---|
| 1 | Smooth flow without resonance | - | 0.701 | 0.632 | Symmetric mode ($r_{sym}$) |
| 2 | Smooth flow with resonance | 0.907 | - | 0.352 | |
| 3 | Gradient flow | 0.547 | 0.670 | - | |
| | | Anti-symmetric mode ($r_{anti}$) | | | |

### 5.3. Wind Pressure Fields of Symmetric Mode

The $j^{th}$ recomposed fluctuating wind pressure field $\boldsymbol{p}_j = \left\{ p_{j1}(t), \cdots, p_{jN}(t) \right\}^T$, represented by mode $\boldsymbol{\phi}_j$ can be expressed using the $j^{th}$ principal coordinate $\widetilde{a}_j(t)$ as follows, where $p_{jk}(t)$ is the $j^{th}$ recomposed fluctuating wind pressure at point $k$:

$$\boldsymbol{p}_j = \mathrm{Re}\left[ \boldsymbol{\phi}_j \widetilde{a}_j(t) \right]. \tag{31}$$

The recomposed fluctuating wind pressure fields were investigated for the 1st- and 2nd-modes using $p_{jk}(t)$. Hereafter, the characteristics of the fluctuating wind pressure fields for each type in Table 2 are shown by comparing the following three measurement cases:

- Typical measurement case of Type 1: Rigid model, smooth flow, $V_r = 9.7$.
- Typical measurement case of Type 2: Elastic model, smooth flow, $V_r = 9.7$, $h = 1\%$
- Typical measurement case of Type 3: Elastic model, gradient flow, $V_r = 9.6$, $h = 1\%$

The recomposed fluctuating wind pressure fields in the symmetric mode for each type are shown in Figure 46. Furthermore, the pressure rise, or recovery region, is shown in red, the pressure drop region in blue, and the intensity of the fluctuating wind pressure is indicated by the shade of the colour. The wind pressure was divided by the velocity pressure.

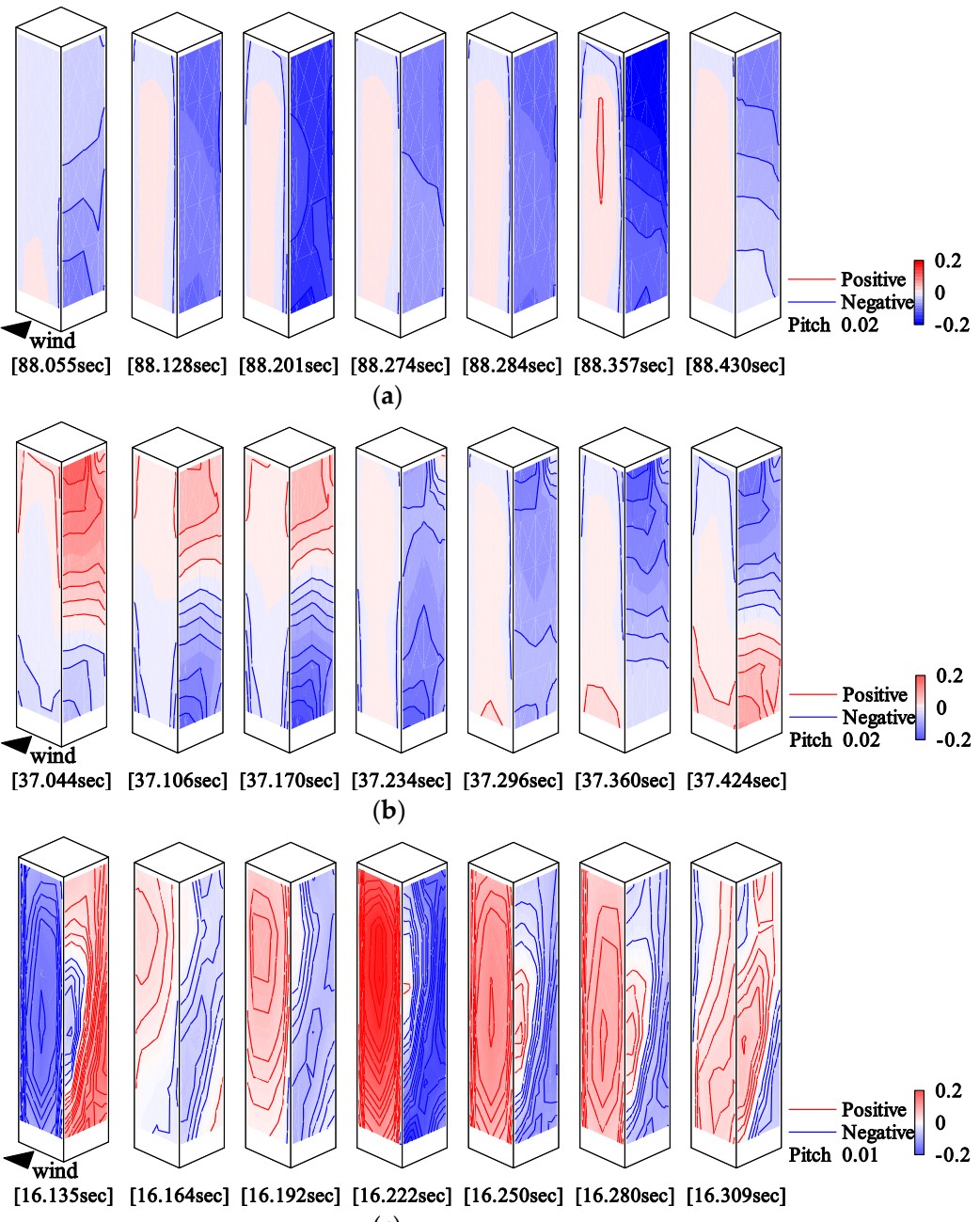

**Figure 46.** Recomposition of fluctuating wind pressure fields (symmetric mode): (**a**) Type 1; (**b**) Type 2; (**c**) Type 3.

For Type 1 and Type 2, on the side surface, the pressure drop region slowly moved from the area near the bottom to the top. The relatively high correlation of $r_{sym,12} = 0.701$ may be because of the similar trends in the fluctuating wind pressure fields on the side surface. For Type 3, on the side surface, a pressure drop region appeared near the height of $2H/3$ on the windward side and moved toward the bottom of the leeward side. In the case of Type 3, shown in Figure 46c, the pressure on the front surface increases simultaneously with the pressure drop on the side surface. For Types 2 and 3, the fluctuating wind pressure fields on the side surface have very different trends, and for this reason the correlation coefficient, $r_{sym,23}$, is likely to be low. However, for Types 1 and 3, the fluctuating wind pressure fields appear to be different on both the front and side surfaces, although $r_{sym,31}$ is relatively high. The reason for this is unknown.

### 5.4. Wind Pressure Fields of Anti-Symmetric Mode

The recomposed fluctuating wind pressure fields of the anti-symmetric mode for each type are shown in Figure 47. For Type 1, the pressure drop region moves toward the leeward edge near the bottom of the model on the side surface. For Type 2, the pressure drop region near the windward edge at a height of approximately H/4 moves toward the leeward edge of the side at a height of approximately 3H/4 on the side surface. The reason for the very high correlation of $r_{anti,12} = 0.907$ is that the pressure drop region exhibits a similar tendency. For Type 3, the pressure drop region near the windward edge of the side at a height of 3H/4 moves to the leeward edge near the bottom of the model on the side surface. Additionally, the pressure rise region appearing at the right edge of the front extends over the entire front surface, as shown from 62.055 to 62.119 s. This is because the relatively low correlation of $r_{anti,31} = 0.547$ and $r_{anti,23} = 0.670$ may be the differences in the fluctuating wind pressure fields on the front surface.

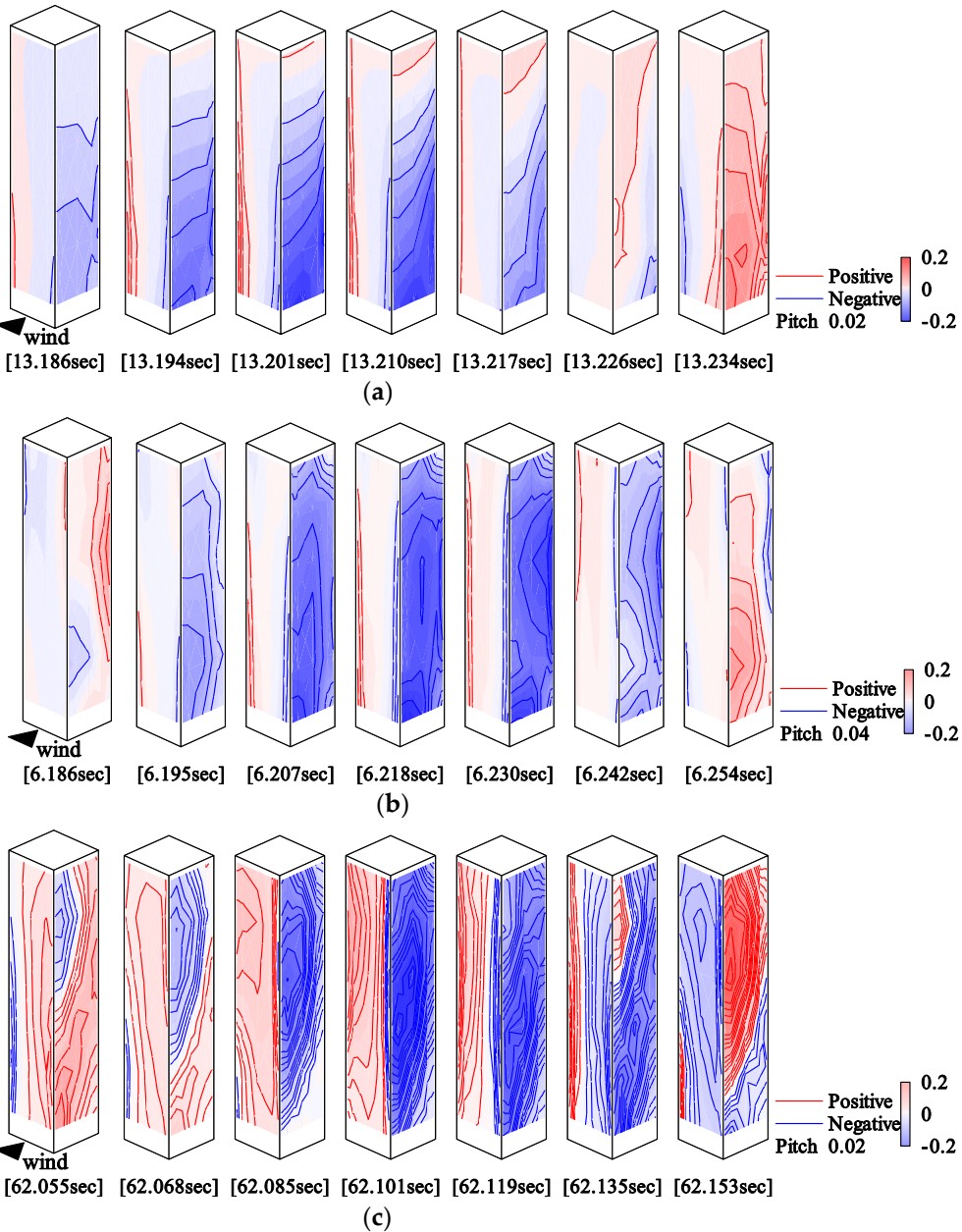

**Figure 47.** Recomposition of fluctuating wind pressure fields (anti-symmetric mode): (**a**) Type 1; (**b**) Type 2; (**c**) Type 3.

## 6. Concluding Remarks

The aim of this study was to evaluate the phase characteristics of fluctuating wind pressure fields around buildings and to provide valuable knowledge for the design of wind-resistant high-rise buildings. For these purposes, wind tunnel experiments were conducted in smooth and gradient flows, measuring the pressures around a rigid and elastic model and the displacement of the elastic model. The characteristics of the wind pressure distribution around the rigid and elastic models were clarified based on the experimental results. In addition, the characteristics of the measured displacement of the elastic model are explained.

A CPOD analysis was performed to investigate the fluctuating wind pressure fields around the prism. The contribution ratio of the anti-symmetric mode for the elastic model with a damping ratio of 1% in the smooth flow of the resonant velocity is much higher than that of the rigid model. Therefore, the fluctuating wind pressure fields around the prism are affected by the vibration of the model. Furthermore, there were no significant differences in the contribution ratios and eigenmodes between the rigid and elastic models in the gradient flow.

A method for evaluating the symmetricity of the eigenmodes was proposed to clarify the phase characteristics of the wind pressure fields around buildings. In the case of the elastic model in the smooth flow, symmetric or anti-symmetric modes are clearly affected by whether the wind speed is in the range of the resonance wind speed or not. In contrast, the symmetricity of the eigenmodes for the rigid model in the smooth flow and for the rigid and elastic models in the gradient flow showed different trends with wind speed. This may be because of the close proximity of the 1st- and 2nd- eigenvalues.

In addition, a method for evaluating the similarity of fluctuating wind pressure fields composed of the 1st- and 2nd- modes as the projection ratio of the plane formed by the two principal eigenmodes was proposed. The results show that fluctuating wind pressure fields can be classified into three types: near the resonant and non-resonant states in smooth flow, and in gradient flow.

The three types of fluctuating wind pressure fields were recomposed to show the phase characteristics of the fluctuating wind pressure fields of the symmetric and anti-symmetric modes. In the case of symmetric modes, the fluctuating wind pressure fields in the smooth flow without the resonance phenomenon tend to be similar to those in the case of smooth flow with resonance and gradient flow. However, the fluctuating wind pressure fields in the smooth flow without resonance and the gradient flow show different patterns. In the case of anti-symmetric modes, the fluctuating wind pressure fields in the case of smooth flow without resonance and with resonance tend to be very close. However, the case of gradient flow shows different patterns. Despite the contribution ratios being different for the rigid and elastic models, the phase characteristics of the fluctuating wind pressure fields for both models are almost the same.

**Author Contributions:** Conceptualization, T.M. and T.T.; methodology, T.M. and T.T.; software, T.T.; validation, T.M., T.T. and Y.N.; formal analysis, T.T. and Y.N.; investigation, T.M.; resources, T.M.; data curation, T.M.; writing—original draft preparation, T.M., T.T. and Y.N.; writing—review and editing, T.M. and T.T.; visualization, T.M., T.T. and Y.N.; supervision, T.T.; project administration, T.T. All authors have read and agreed to the published version of the manuscript.

**Funding:** This research received no external funding.

**Institutional Review Board Statement:** Not applicable.

**Informed Consent Statement:** Not applicable.

**Conflicts of Interest:** The authors declare no conflict of interest.

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
