# Peer review of "Study on Phase Characteristics of Wind Pressure Fields around a Prism Using Complex Proper Orthogonal Decomposition"

_2674-032X, doi:10.3390/wind3010004_

Round 1

Reviewer 1 Report

This is a very good paper. It discussed the phase characteristics of the surface pressure of rigid/elastic prism models in uniform and boundary layer flows. The results are very important for wind-resistant design of high-rise buildings.

A small question: in section 2.2 for rigid model, the reduced velocity is defined as Vr=UH/f0B and f0 is the natural frequency f0=8.34Hz. What is the natural frequency of a rigid model? the natural frequency of the elastic model?

Reviewer 2 Report

This paper is believed to provide valuable knowledge for the wind-resistant design of tall buildings.

But the authors are advised to add more information on the evaluation of damping ratio. Based on the previous study, the damping ratios become very large and show clear amplitude dependency when there are pressure tubes on vibrating model. The authors need to show more clearly 1) how to evaluate damping ratios, 2) characteristics of amplitude dependency of damping ratios, 3) effect of damping ratios on the CPOD results, and 4) compare the current results with the ones in the previous study, i.e. Kim et al., 2018, JWEIA.

Round 2

Reviewer 2 Report

The manuscript was revised and improved properly.